# Functional Wasserstein Variational Policy Optimization

**Junyu Xuan**[1]          **Mengjing Wu**[1]          **Zihe Liu**[1]          **Jie Lu**[1]

[1]Australian Artificial Intelligence Institute, University of Technology Sydney, Ultimo NSW 2007, Australia,

## Abstract

Variational policy optimization has become increasingly attractive to the reinforcement learning community because of its strong capability in uncertainty modeling and environment generalization. However, almost all existing studies in this area rely on Kullback–Leibler (KL) divergence which is unfortunately ill-defined in several situations. In addition, the policy is parameterized and optimized in weight space, which may not only bring additional unnecessary bias but also make the policy learning harder due to the complicatedly dependent weight posterior. In the paper, we design a novel functional Wasserstein variational policy optimization (FWVPO) based on the Wasserstein distance between function distributions. Specifically, we firstly parameterize policy as a Bayesian neural network but from a function-space view rather than a weight-space view and then propose FWVPO to optimize and explore the functional policy posterior. We prove that our FWVPO is a valid variational Bayesian objective and also guarantees the monotonic expected reward improvement under certain conditions. Experimental results on multiple reinforcement learning tasks demonstrate the efficiency of our new algorithm in terms of both cumulative rewards and uncertainty modeling capability.

## 1 INTRODUCTION

Reinforcement learning aims to optimize a policy that could yield high cumulative rewards when interacting with a given environment. One straightforward solution is to parameterize the policy, represent the cumulative reward as a function of the policy, and maximize the cumulative reward by optimizing the policy. Such a solution is named policy optimiza-tion[1] [Schulman et al., 2015, 2017, Huang et al., 2021] or policy gradient [Williams, 1992, Li et al., 2021, Castellini et al., 2021]. Popular algorithms include trust region policy optimization (TRPO) [Schulman et al., 2015], proximal policy optimization (PPO) [Schulman et al., 2017], Bregman gradient policy optimization [Huang et al., 2021], and so on. Almost all of these works parameterize the policy as a determinate deep neural network in which capability in uncertainty modeling and environment generalizing is limited [Furmston and Barber, 2010].

One way to improve the ability of uncertainty modeling and environment generalizing is to parameterize the policy as a probabilistic model [Furmston and Barber, 2010, Levine, 2018, Xu, 2018]. Among all possible probabilistic models, Bayesian neural networks (BNNs) [Blundell et al., 2015, Foong et al., 2020], which assign probabilistic distributions on all weights of the neural networks are one of the most popular options because they absorb the advantages of deep neural networks on the powerful function approximation. The underlying reason for this parameterization is that it can transform the reinforcement learning as a probabilistic inference problem and then various approximate probabilistic inference algorithms can be used to provide additional flexibility and representation power [Levine, 2018, Zhang et al., 2020, 2018] and effective reasoning about uncertainty [Fellows et al., 2019, Liu et al., 2017]. Hence, variational inference [Blei et al., 2017] has been broadly used to improve the policy optimization (named variational policy optimization), where a Kullback–Leibler (KL) divergence is added to constrain the posterior distribution of the policy. One representative work is the maximum entropy policy optimization (MEPO) [Levine, 2018, Liu et al., 2017].

Unfortunately, there is no such thing as a free lunch. Introducing BNN and variational inference to policy optimization also brings additional difficulties, like i) the widely used

---

Gaussian priors for network parameters are not always applicable due to their possible pathological features, such as prior samples tend to be horizontally linear for deep nets [Duvenaud et al., 2014, Tran et al., 2020]; ii) the effects of the given priors on posterior inference for weights and further on the resulting distributions over model outputs in function space are unclear and hard to control owing to the complex architecture and nonlinear nature of BNNs [Ma and Hernández-Lobato, 2021, Wild et al., 2022]. Both these difficulties source from the independent distributed prior and strong and complicatedly dependent posterior of policy network weights.

In this paper, instead of parameterizing policy in weight space, we propose a functional variational policy optimization algorithm, where a policy is given a functional prior and its posterior is optimized in function space [Rasmussen and Williams, 2006]. Although there are some recent ingenuous works on functional variational inference for BNNs [Sun et al., 2019, Wang et al., 2018, Ma and Hernández-Lobato, 2021], they are all based on KL divergence which has natural relationship with data log likelihood (the variational objective is a lower bound of data log-likelihood) but is either infinite or ill-defined in several situations [Gray, 2011, Burt et al., 2020], like non-overlapping supports. Moreover, the KL divergence is known vulnerable to collapse to local mode [Neumann et al., 2011] and hence sensitive to the initialization (please see Section 3 for more discussions). Therefore, when the existing functional variational inference with KL divergence is directly used as the surrogate objective function for policy optimization, it could be harmful to the monotonic improvement of each step and may lead to instability. Our basic idea in a nutshell is to use Wasserstein distance [Arjovsky et al., 2017, Ambrogioni et al., 2018] between policy posterior and prior as the constraint and use functional Wasserstein variational inference as the surrogate objective function for policy optimization. Our main contributions are summarised as follows,

- We propose a new functional Wasserstein variational inference based on 1-Wasserstein distance rather than KL divergence, where the new objective is proven to be a valid and tighter (compared with KL) variational Bayesian objective.
- We derive a functional Wasserstein variational policy optimization (FWVPO), prove the monotonic improvement guarantee and demonstrate the improvement compared with KL divergence.

## 2 BACKGROUND

### 2.1 REINFORCEMENT LEARNING

Reinforcement learning can be formalized by Markov decision processes (MDP). A MDP is defined by a tuple $\{\mathcal{S}, \nu, \mathcal{A}, R, P\}$, where $\mathcal{S}$ is the state space, $\nu$ is the starting distribution of states, $\mathcal{A}$ is the action space, $R(r|s,a)$ is the reward function $R : \mathcal{S} \times \mathcal{A} \rightarrow \mathbb{R}$, and $P(s_{t+1}|s_t, a)$ is the state transition probability. A policy $\pi(a|s; \theta)$ is a distribution over actions given a state, with $\theta$ as the parameter set. When a deep neural network is used to model $\pi_\theta$, $\theta$ contains the weights of the network. With discount factor $\gamma \in (0, 1)$, the expected discounted reward under $\pi_\theta$ is defined as

$$\eta(\pi_\theta) = \mathbb{E}_{s_0, a_0, \ldots} \left[ \sum_{t=0}^{\infty} \gamma^t R(s_t, a_t) \right] \quad (1)$$

where $s_0 \sim \nu, a_0 \sim \pi(s_0), s_1 \sim P(s_0, a_0), \ldots$. The definitions of standard concepts, including state-action value function $Q_\pi(s, a)$, state value function $V_\pi(s)$, and the advantage function $A_\pi(s, a)$, follow the ones in TRPO [Schulman et al., 2015] and are given in the Supplementary.

### 2.2 POLICY OPTIMIZATION

The aim of policy optimization algorithms [Williams, 1992] is to maximize the expected discounted reward in (1) by optimizing the policy parameters. One problem of the standard policy gradient is the possible collapse due to a large update step. TRPO [Schulman et al., 2015] nicely avoids this kind of collapse through a KL divergence between the old and new policies that are given to restricting the update, and a ratio to compensate the difference between trajectory collecting (old) policy $\pi_{\theta_{\text{old}}}$ and current policy $\pi_\theta$ by

$$\max_\theta \quad J^{\text{TRPO}}(\theta) - \alpha \mathcal{KL} \left[ \pi_{\theta_{\text{old}}} \| \pi_\theta \right] \quad (2)$$

where $\alpha$ is a hyperparameter and $J^{\text{TRPO}}(\theta) = \mathbb{E}_{s_0, a_0, \ldots} \left[ \frac{\pi(a|s; \theta)}{\pi(a|s; \theta_{\text{old}})} A_{\pi_{\theta_{\text{old}}}}(s, a) \right]$. PPO [Schulman et al., 2017] further extends TRPO by introducing a clipped surrogate that maximizes the cost function while ensuring the deviation from the previous policy is relatively small $J^{\text{PPO}}(\theta) = \mathbb{E}_{s_0, a_0, \ldots} \left[ \min(\frac{\pi(a|s; \theta)}{\pi(a|s; \theta_{\text{old}})} A_{\pi_{\theta_{\text{old}}}}(s, a), \text{clip}(\frac{\pi(a|s; \theta)}{\pi(a|s; \theta_{\text{old}})}, 1 - \epsilon, 1 + \epsilon) A_{\pi_{\theta_{\text{old}}}}(s, a)) \right]$ where $\epsilon$ is a hyperparameter. Variational policy optimization (like MEPO) [Levine, 2018, Liu et al., 2017] is to introduce a policy prior distribution $p_0(\theta)$ and the target is to optimize the approximated policy posterior distribution $q(\theta)$ by $\max_q \mathbb{E}_{q(\theta)} [J(\theta)] - \alpha \mathcal{KL}[q(\theta) \| p_0(\theta)]$, where $\alpha$ is hyperparameter and $J(\theta)$ can be any surrogate term, like $J^{\text{TRPO}}(\theta)$ or $J^{\text{PPO}}(\theta)$, and the optimal posterior can be directly deduced as $q^*(\theta) \propto \exp(J(\theta)) p_0(\theta)$.

### 2.3 BAYESIAN NEURAL NETWORKS

A Bayesian neural network (BNN) [Blundell et al., 2015, Foong et al., 2020] is a neural network whose weights are

given (normally independent) prior distributions. Given a dataset $\mathcal{D} = \{x_i, y_i\}$ where $x_i \in \mathbb{R}^d$ and $y_i \in \mathbb{R}$, a simple one-hidden-layer example is given as $y(x) = \theta^{(0)} + \theta^{(1)}\sigma\left(\theta^{(2)} + \theta^{(3)}x\right)$, where $\sigma$ is a nonlinear activation function, and $\theta = \{\theta^{(0)}, \theta^{(1)}, \theta^{(2)}, \theta^{(3)}\}$ are neural network weights with prior distribution $p_0(\theta)$, such as i.i.d. Gaussian distributions. There are various approximate inference methods to optimize their complex posterior distributions, such as variational inference (VI) [Blundell et al., 2015] and Hamiltonian Monte Carlo [Cobb and Jalaian, 2021]. Here, we briefly introduce the mean-field VI for BNN, which proposes some simple (like Gaussian) independent variational distributions $q(\theta; \vartheta)$ with $\theta \sim \text{Gaussian}(\bar{\mu}, \bar{\rho})$, where $\vartheta = \{\bar{\mu}, \bar{\rho}\}$ is also named as the variational parameter, which is trained to closely approximate the true posterior distribution. For data $\mathcal{D}$, the loss function is

$$\max_{\vartheta} \mathbb{E}_{q(\theta;\vartheta)}\left[\sum_i \log p(y_i|x_i;\theta)\right] - \alpha\mathcal{KL}\left[q(\theta;\vartheta)\|p_0(\theta)\right] \tag{3}$$

where $p(y_i|x_i;\theta)$ is the data likelihood and could be a categorical distribution for classification task or Gaussian distribution for regression; the first is also known as expected log-likelihood, which variance could be further reduced by local reparameterization trick [Kingma et al., 2015].

# 3 FUNCTIONAL WASSERSTEIN VARIATIONAL POLICY OPTIMIZATION

The policy is traditionally parameterized by a deterministic deep neural network in which the final layer outputs parameters of an (action) distribution $\varpi(a|s;\theta)$. Although an action distribution is learned, such a design has limited capability to capture the uncertainty of this distribution because of the deterministic structure of the neural network. To resolve such an issue, BNN was used to replace the deterministic deep neural network [Levine, 2018, Liu et al., 2017], i.e., $\pi(a|s) = \mathbb{E}_{p(\theta;\vartheta)}[\varpi(a|s;\theta)]$. However, BNN is only used in weight space by the existing works, which greatly reduces the ability in function flexibility (due to space limitation, more details about the difference between weight-space and function-space can be found in [Rasmussen and Williams, 2006]). Hence, we use BNN as the policy representation but work in the function space rather than the weight space, i.e., $\pi(a|s) = \mathbb{E}_{p(f)}[\varpi(a|f(s))] = \mathbb{E}_{p(\theta^f;\vartheta^f)}[\varpi(a|s;\theta^f)]$, where $p(f)$ is a functional distribution induced by a parameterized BNN with $p(\theta^f;\vartheta^f)$. In a nutshell, $p(f)$ can be simply understood as a BNN whose weights are with a distribution parameterized by $\vartheta^f$. More details about the differences between deterministic policy, policy parameterized by BNN in weight space, and policy parameterized by BNN in function space are given in the Supplementary.

Inspired by the existing functional BNNs [Sun et al., 2019,

Wang et al., 2018, Ma and Hernández-Lobato, 2021], we have the following initial functional variational policy optimization (FVPO),

$$\max_{q} \quad \mathbb{E}_{q(f)}\left[J(f)\right] - \alpha\mathcal{KL}\left[q(f)\|p_0(f)\right] \tag{4}$$

where $f$ is a policy function (mapping from state to action); $p_0(f)$ is a functional prior, such as Gaussian process [Rasmussen and Williams, 2006]; similar with $q(\theta;\vartheta)$ in (3), $q(f)$ is an approximated functional posterior induced by a parameterized BNN with $q(\theta^f;\vartheta^f)$; and $J(f)$ is the surrogate term and can be evaluated as $\mathbb{E}_{q(f)}[J(f)] = \mathbb{E}_{q(\theta^f;\vartheta^f)}[J(\theta^f)]$ and $J(\theta^f)$ can be $J^{\text{TRPO}}(\theta^f)$ or $J^{\text{PPO}}(\theta^f)$.

The first term of FVPO is ordinary so we are more interested in the second functional KL divergence term. Before investigating this functional KL divergence, let us first look at the merits of this functional policy optimization: 1) the optimal function posterior can be directly deduced as $q^*(f) \propto \exp\left(J(f)\right)p_0(f)$, but it is unfortunate that we normally do not have an explicit function probability density form to express such posterior easily; and 2) optimizing KL divergence between function distributions is hard but doable because there is a link with its marginal KL divergence on measurement set [Sun et al., 2019, Gray, 2011] as demonstrated by Theorem 1 in [Sun et al., 2019] that is $\mathcal{KL}[P\|Q] = \sup_{n\in\mathcal{N}, X\in\mathcal{X}^n} \mathcal{KL}[P_X\|Q_X]$ where $P$ and $Q$ are two stochastic processes defined on space $\mathcal{X}$ and $P_X$ and $Q_X$ are their marginals on $X$ respectively.

Although FVPO is a great initial effort to transform to function space variational inference, unfortunately, this functional KL divergence may be an ill-defined objective function sometimes.

**Definition 1** (Functional KL divergence [Gray, 2011]). Suppose we have two measures $P$ and $Q$ for $(\Omega, \Sigma)$ and that $P$ is absolutely continuous with respect to $Q$. Then there exists a Radon-Nikodyn derivative $\text{dP}/\text{dQ}$ and the KL-divergence between them is $\mathcal{KL}[P\|Q] = \int_\Omega \log\{\text{dP}/\text{dQ}\}\,\text{dP}$.

According to the above definition, $\mathcal{KL}[P\|Q] = \infty$ if $P$ is not absolutely continuous with respect to $Q$. Besides, Burt et al. [2020] also found that $\mathcal{KL}[P\|Q] = \infty$ if the network architectures of prior and approximated posterior are different or prior is a non-degenerate Gaussian process.

To resolve this issue, we propose to use Wasserstein distance to replace KL divergence, and then we have the following functional Wasserstein variational policy optimization (FWVPO),

$$\max_{q} \quad \mathbb{E}_{q(f)}\left[J(f)\right] - \left(\mathcal{W}[q(f)\|p_0(f)]\right)^2 \tag{5}$$

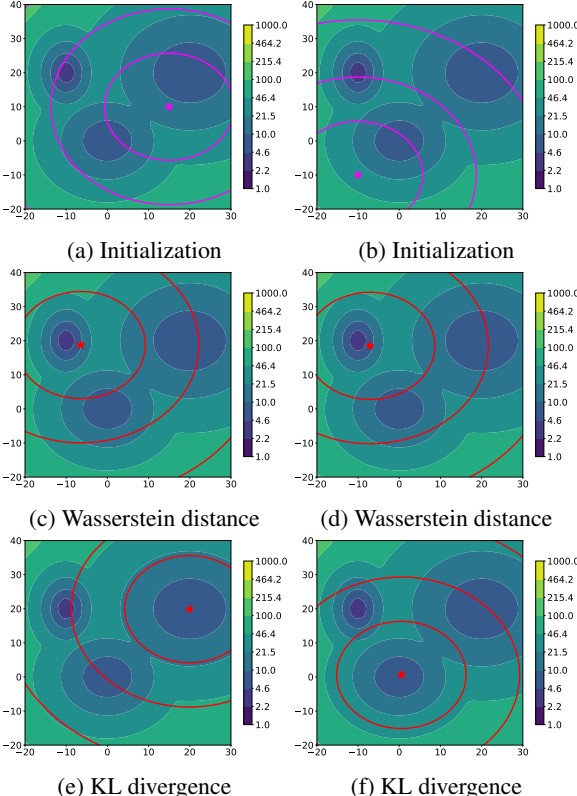

(a) Initialization      (b) Initialization

(c) Wasserstein distance      (d) Wasserstein distance

(e) KL divergence      (f) KL divergence

Figure 1: Approximation results from different loss. The background contour field is a Gaussian mixture with three components. A single-mode Gaussian distribution is optimized to approximate this mixture distribution using loss defined by three metrics. The initial position of this Gaussian distribution is given in (a) and the approximation results are plotted with contour (red) lines. The detailed setting is given in the Supplementary.

where $\mathcal{W}$ denotes 1-Wasserstein distance[2]

$$\mathcal{W}[q(f)\|p_0(f)] = \inf_{p(f,f')} \int c(f, f')p(f, f')\mathrm{d}f\mathrm{d}f' \quad (6)$$

and $p(f, f')$ is any joint distribution with $f \sim q(f)$ and $f' \sim p_0(f)$ as marginals and $c$ is a cost function (metric). It is worth highlighting that (5) is a kind of generalized variational inference (or more general *Rule of Three*) [Knoblauch et al., 2019] rather than a standard Bayesian inference objective because the loss function term $\mathbb{E}_{q(f)}[J(f)]$ is not a conditional distribution and Wasserstein distance is used instead of KL divergence. It is a well-defined distance measure in terms of being positive, symmetric and well-behaved in the situations [Ambrogioni et al., 2018, Arras et al., 2019] where KL may be infinite or unbounded due to Radon-Nikodyn derivative [Matthews et al., 2016]. Note that different from

---

[2]We only use 1-Wasserstein distance throughout this paper, so $\mathcal{W}$ for the remainder of the paper denotes 1-Wasserstein distance without further notice.

[Ambrogioni et al., 2018] where the Wasserstein distance is defined on parameter distributions, the distance in (5) is defined on the function distributions; different from [Wild et al., 2022] where the 2-Wasserstein distance is used to define a functional variational inference, (5) does not simplify the BNN posterior to be a Gaussian process (parameterized by a neural network mean function) which is adopted in [Wild et al., 2022] and limits the uncertainty representation capability.

As also argued in [Neumann et al., 2011] and shown in Figure 1, KL divergence could concentrate at a mode of the target distribution when it is a multi-modal or non-concave target policy distribution. However, we argue that KL divergence may collapse at a local mode so is sensitive to the initialization, but the Wasserstein distance could jump out of the local mode to find a better/higher one as shown in Figure 1. This is desirable to RL because we hope to search for the best policy during the update rather than collapsing in the local optimum.

To facilitate the optimization, we use its dual form according to the Kantorovich-Rubinstein duality [Villani, 2009, Arjovsky et al., 2017],

$$\mathcal{W}[q(f)\|p_0(f)] = \max_{\|\phi\|_{L\leq 1}} \mathbb{E}_{q(f)}[\phi(f)] - \mathbb{E}_{p_0(f)}[\phi(f)] \quad (7)$$

where $\|\phi\|_{L\leq 1}$ denotes that $\phi$ is constrained to 1-Lipschitz function. This duality can nicely separate two marginal distributions and the evaluation can be achieved through sampling-based methods. A similar idea is also used for prior matching [Tran et al., 2022] but we use it for the posterior variational inference here. There are various ways [Tanielian and Biau, 2021, Gulrajani et al., 2017] to approximate a 1-Lipschitz function as a deep neural network to facilitate the (sub)optimal function searching. Here, we use a gradient norm regularizer to ensure $\phi$ is a 1-Lipschitz function [Tran et al., 2022, Gulrajani et al., 2017] and then search the space to find the one to maximize (7).

Note that any variational Bayesian methods [Fox and Roberts, 2012] need to derive a lower bound for the marginal data likelihood (also known as evidence) as the model training objective function. One natural and important question is: Will (5) still be a valid variational objective? In short, can we use "variational" here? We answer this question with the following result.

**Theorem 1.** *Let $p_0(f)$ be a function prior (like the Gaussian process) and parameterize the policy as $\pi(a|s) = \mathbb{E}_{f\sim q(f)}[\varpi(\cdot|f, s)]$ where $q(f)$ is a function distribution induced by BNN weight distributions. Given a measurement set $S$, if $-\log p_0(f^S)$ is a Lipschitz function and probability measure $p_0$ absolutely continuous w.r.t. $q$,*

$$\log p(D) \geq \mathcal{L}^{\mathcal{W}} \geq \mathcal{L}^{\mathcal{KL}}$$

*where*

$$\mathcal{L}^{\mathcal{W}} = \mathbb{E}_{q(f)}\left[J(f)\right] - \frac{\rho}{2}\left(\mathcal{W}[q(f^S)\|p_0(f^S)]\right)^2,$$
$$s.t., H(q) - H(p_0) - \frac{1}{2\rho} \geq 0; \rho > 0 \tag{8}$$

*and*

$$\mathcal{L}^{\mathcal{KL}} = \mathbb{E}_{q(f)}\left[J(f)\right] - \mathcal{KL}[q(f^S)\|p_0(f^S)].$$

*and* $p(D|f) \propto \exp(J(D,f))$, $\log p(D) = \log\left(\int_f p(D|f)\mathrm{d}f\right)$.

*Proof.* Please see the Supplementary. □

Theorem 1 confirms that (5) is a lower bound for the marginal data likelihood so it is a valid variational objective and a tighter bound compared with KL divergence.

The other question is: Can (5) still hold the monotonic improvement guarantee like TRPO and is there any improvement compared with KL divergence? We answer this question with the following result.

**Theorem 2.** *Let an old policy (before a training step) is parameterized by a BNN with function prior $p(f)$, i.e., $\pi(a|s) = \mathbb{E}_{f\sim p(f)}\left[\varpi(a|s;\theta^f)\right]$, a new policy is parameterized by a BNN with function prior $\tilde{p}(f)$, i.e., $\tilde{\pi}(a|s) = \mathbb{E}_{f\sim\tilde{p}(f)}\left[\varpi(a|s;\theta^f)\right]$, $\varpi(a|s;\theta)$ defines an action distribution by a (deterministic) neural network parameterized by $\theta$, and $L_\pi(\tilde{\pi})$ is the expected reward of $\tilde{\pi}$ evaluated on the trajectory generated by $\pi$. if $0 < ||\tilde{\pi}||_1 < \infty, 0 < ||\pi||_1 < \infty$, then the following bound holds*

$$\eta(\tilde{\pi}) \geq L_\pi(\tilde{\pi}) - \frac{1}{1-\gamma}\left(\mathcal{W}\left[\tilde{p}(f)\|p(f)\right]\right)^2 \tag{9}$$

*where $\mathcal{W}^{max}\left[\tilde{p}(f)\|p(f)\right] = \max_s \mathcal{W}\left[\tilde{p}(f)\|p(f)\right]$ and the equality holds when $\tilde{p}(f) = p(f)$. Moreover,*

$$\eta_{\mathcal{W}} \geq \eta_{\mathcal{KL}} \tag{10}$$

*where $\eta_{KL} = L_\pi(\tilde{\pi}) - \frac{1}{1-\gamma}\mathcal{KL}\left[\tilde{p}(f)\|p(f)\right]$.*

*Proof.* We provide the proof in the Supplementary, where we use the relationship between total variation divergence, KL divergence and Wasserstein distance. □

Theorem 2 states that the optimization of the right-hand side (RHS) of (9) can guarantee the monotonic improvement of the expected reward. The RHS of Theorem 2 just corresponds to the objective function in (5). The only difference is that the distance is between the prior and posterior in (5) but is between two consecutive posteriors in (9). We can understand (5) as the initial constraint with no other previous knowledge and (9) as the continual constraint using updated

knowledge about the posterior, or we can also understand (5) as the global constraint and (9) is the local constraint.

Another point we need to highlight is that we hope to preserve the stochastic process properties (i.e., marginalization consistency according to Kolmogorov Extension Theorem [Øksendal, 2003]) of $q(f)$ during the optimization because the marginalization consistency could greatly improve the generalization ability of the learned policy. However, the approximation of using function samples and finite measurement sets may damage such properties. To further ensure the marginalization consistency of $q(f)$, we then propose to minimize the distance between the marginal $q_j(f(Y))$ of a joint distribution $q(f(Y,U))$ using samples $(Y,U)$ and $q_m(f(Y))$ only using samples $Y$ by

$$\mathcal{W}_Y\left[q_m(f(Y)), q_j(f(Y))\right] =$$
$$\max_{\|\phi\|_L\leq 1}\left|\frac{1}{N_j}\sum_{i=1:N_j}\phi(f_{j,i}(Y)) - \frac{1}{N_m}\sum_{i=1:N_m}\phi(f_{m,i}(Y))\right| \tag{11}$$

where $f_{j,i}$ and $f_{m,i}$ are both function samples from $q(f)$ but with no overlap and we use subscripts to distinguish them. Since we have the samples of the joint distribution $q(f(Y,U))$, it is easy to obtain its marginal on $Y$ by just throwing $U$ away. Ideally, all possible $Y$ would be better evaluated using the above formula, but the combinatorial number is too large, so we only (uniformly) randomly sample several sets instead.

To sum up, we integrate the three terms gradually proposed above to obtain our final FWVPO,

$$\max_q \quad \mathbb{E}_{q(f)}\left[J(f)\right] - \alpha_1\left(\mathcal{W}\left[q_{old}(f)\|q(f)\right]\right)^2$$
$$- \alpha_2\left(\mathcal{W}[q(f)\|p_0(f)]\right)^2$$
$$- \alpha_3\mathcal{W}_Y\left[q_m(f(Y)), q_j(f(Y))\right] \tag{12}$$
$$s.t. \quad H(q) - H(p_0) - \frac{1}{2\rho} \geq 0$$

where $\{\alpha > 0\}$ and $\rho > 0$ are hyperparameters. The first regularizer corresponds to the monotonic improvement property from (9); the second regularizer corresponds to the prior constraint from (8); the last is to enhance marginalization consistency from (11). It is interesting to see that the prior can be considered as a *global* and *static* constraint while the $q_{old}$ can be considered as a *local* and *dynamic* constraint. In practice, we use a finite measurement set $S \in \mathcal{S}^k$ to evaluate the above objective functions (12) according to (7): $\mathbb{E}_{q(f)}\left[\phi(f(S))\right] - \mathbb{E}_{p_0(f)}\left[\phi(f(S))\right]$ where $k$ is size of the measurement set. For each training step, we first add a batch of training episodes in local on-policy memory buffer to the measurement set and randomly select a set from a global pool that stores all visited states. The core part is summarised in Algorithm 1 (see Supplementary).

# 4 RELATED WORK

Policy optimization algorithms can be roughly grouped according to gradient and model usage: gradient-based methods [Williams, 1992, Li et al., 2021, Castellini et al., 2021] and gradient-free methods [Szita and Lörincz, 2006]; model-based and model-free methods. The focus of this paper is on model-free gradient-based policy optimization only. Inspired by the conservative policy iteration [Kakade and Langford, 2002], the TRPO [Schulman et al., 2015] and PPO [Schulman et al., 2017] were proposed to generalize the idea to general stochastic policies and obtained the state-of-the-art performance. Following these studies, several ingenious ideas were proposed, such as a new clipping function to improve the performance stability [Wang et al., 2020], an additional estimate of the expected return given a policy parameter using Gaussian process to encourage exploration [Rao et al., 2020], convergence analysis of policy optimization algorithms using mirror descent iteration and momentum techniques [Huang et al., 2021], and so on. Variational policy optimization is an interesting branch that borrows the approximate Bayesian inference techniques to improve the uncertainty modeling and generalization capability. One popular approximate Bayesian inference is variational inference, such as MEPO [Levine, 2018, Liu et al., 2017], which naturally transformed the probabilistic inference as an optimization problem with an additional KL divergence between approximate posterior and prior. Since the KL divergence is not symmetric, its reversed version [Neumann et al., 2011] was also used to force the policy to be 'cost-averse' rather than 'reward-attracted' but lost the original lower bound property. Apart from variational inference [Blei et al., 2017], particle filtering was also used to develop the Stein variational policy gradient [Liu et al., 2017] to directly minimize the KL divergence between the optimal posterior distribution and prior through a series of iterative transformed approximate distributions. Another similar idea [Xu, 2018] used amortized variational inference to resolve KL divergence through a general invertible transformation.

Apart from KL divergence, there are also studies trying to use the Wasserstein distance for policy optimization [Terpin et al., 2022]. Such distance has been used to constrain the distance between before and after transition probabilities [Abdullah et al., 2019] and the before and after return distributions [Li and Faisal, 2021]. For policy distance regularization, Wasserstein distance was used to measure the distance between behavioral policy embeddings which encodes global behaviors of policy rather than local action selection [Pacchiano et al., 2020]. To facilitate the policy optimization under Wasserstein regularization, one idea was to link policy optimization with Wasserstein gradient flow, and then a particle approximation method was proposed to estimate such Wasserstein gradient flow [Zhang et al., 2018]; another idea was to use the second-order Taylor ex-

pansion of Wasserstein distance to characterize the local behavioral structures [Moskovitz et al., 2020]; the latest idea [Song et al., 2022] was to derive a closed-form of the policy update based on the Lagrangian of the constrained optimization. Apart from policy optimization, Wasserstein distance was also used for other reinforcement learning tasks, such as reward function learning [Zhang et al., 2019]. Note that although the Wasserstein gradient flow used in [Zhang et al., 2019] can be considered a kind of functional optimization/sampling on the probability measure space, the samples obtained from this reference are still within parameter space. Extending it to the function space requires specific designs [Wang et al., 2018]. Most of these works were in the parameter space rather than the function space considered in our work.

# 5 EXPERIMENTS

We designed our experiments to investigate the following questions: 1) How do different policy parameterizations and prior-posterior distance choices affect the performance of the algorithm? 2) what is the advantage of modeling uncertainty using function distribution? Our code[3] is released for reference.

## 5.1 BASIC SETUP

We used **PPO** as the base model and its clipped objective term $J(f)$ for all comparative methods. We then implemented the policy optimization with BNN as the policy parameterization **BNN-PPO** and its extensions with KL divergence **BNN-KL-PPO** and Wasserstein distance **BNN-W-PPO** [Pacchiano et al., 2020, Zhang et al., 2018] between action distributions, respectively. We also implemented the policy optimization with functional BNN as the policy parameterization and its extensions with KL divergence **fBNN-KL-PPO (FKVPO)** and Wasserstein distance **fBNN-W-PPO (FWVPO)** between function distributions, respectively. All algorithms are given the same hyperparameters whose details can be found in the Supplementary.

## 5.2 EFFECT OF DIFFERENT POLICY PARAMETERIZATIONS AND PRIOR-POSTERIOR DISTANCE CHOICES

The learning curves of algorithms on classical gym environments are shown in Figures 5 and 2, where the x-axis denotes time steps and the y-axis denotes the average episode reward. From Figure 5, we can see that the PPO quickly converged after a small number of steps but became unstable along the training after 2e6 steps. The possible reason is that the PPO fell into a local minimum so its performance

---

[3]https://github.com/JunyuXuan/FWVPO

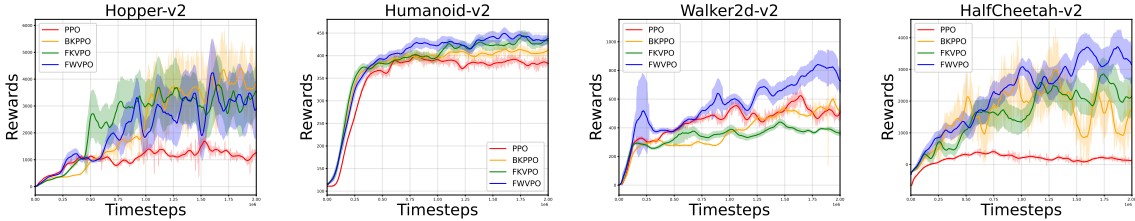

Figure 2: Average episode rewards on four MuJoCo environments.

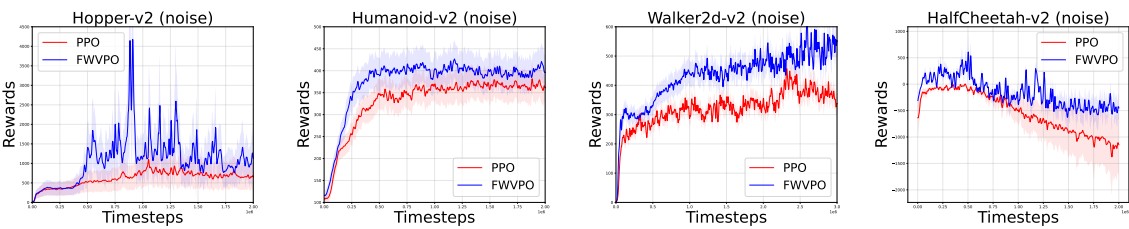

Figure 3: Average episode rewards on four MuJoCo environments with noises.

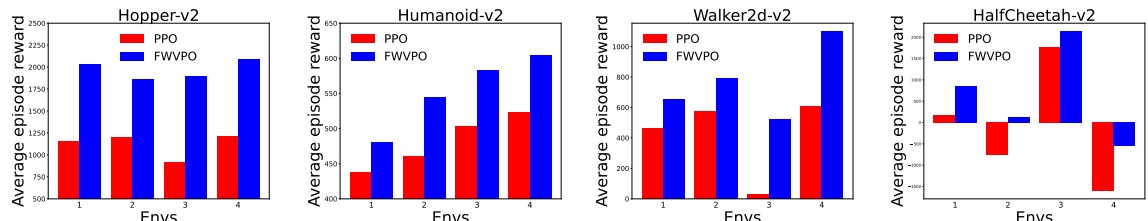

Figure 4: Average episode rewards on four MuJoCo environments under four variations.

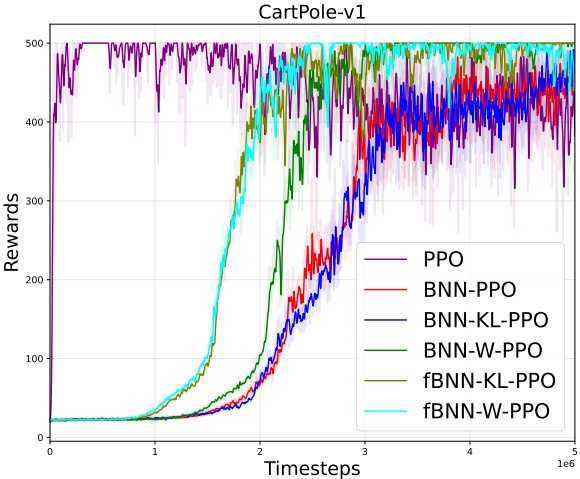

Figure 5: Average episode rewards from different algorithms on CarPole.

dropped after exploring to new state space. Compared with PPO, BNN-PPO had a slow convergence rate in Figure 5. The reason for that is the BNN parameterized policy that will learn a distribution of the functions rather than a single function by a deterministic neural network used in PPO,

which apparently needs more samples. In the more complex MuJoCo environments, the convergence rate is not worse than PPO as shown in Figure 2. The merits of such distribution learning will be demonstrated in the following sections. We observed that BNN-W-PPO was better than both of them due to the Wasserstein distance as we expected. Two functional BNN-based algorithms achieved better performances than all parametric BNN-based ones, where fBNN-W-PPO was slightly better than fBNN-KL-PPO. The reason for that is that we used grid KL divergence [Ma and Hernández-Lobato, 2021] between function distributions which was proven to be bounded.

### 5.3 ROBUSTNESS TO NOISY OBSERVATIONS

One merit of uncertainty modeling[4] is the ability to handle noises. To verify this, we injected (multivariate Gaussian distributed) random noises into an environment (the details of the setup can be found in the Supplementary). The results of PPO and FWVPO on noisy environments are given in

---

[4]There are some works on decomposing aleatoric and epistemic uncertainties for specific tasks, but we did not decompose two kinds of uncertainty and only focused on the general and basic policy optimization task.

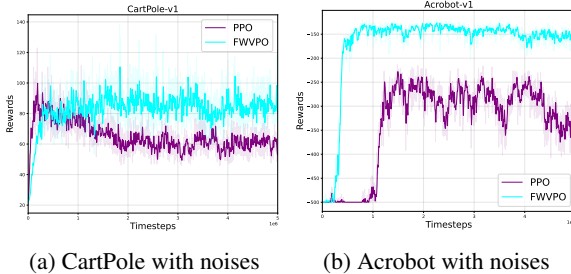

(a) CartPole with noises     (b) Acrobot with noises

Figure 6: Rewards on two noisy environments

Figures 6 and 3. We can see that the rewards from PPO dramatically dropped from -100 (reward from no-noise Acrobat) to -300 (noisy Acrobat) and 500 (reward from no-noise CarPole) to 60 (noisy CarPole) as shown in Figure 6b. On MuJoCo environments, we can also observe that our FWVPO could obtain consistently higher rewards when facing the same noises as PPO in all four environments as shown in Figure 3. It not only converged to a higher reward than PPO but also obtained a faster convergence rate on Acrobot and a comparable rate on CartPole and MuJoCo. We need to highlight that, unlike weakly supervised learning studies [Zhou, 2018], there was no specific component or strategy designed in FWVPO for noises. Such ability is totally from the uncertainty modeling ability.

## 5.4 GENERALIZATION TO ENVIRONMENT VARIATIONS

The other merit of uncertainty modeling is the ability to generalize to environmental variations. To verify this, we will test the pre-trained algorithms on variated environments without further training. The average results on ten episodes in CarPole are shown in Figure 7 where the x-axis denoted the revised 5 environments with different change degrees compared to the basic CarPole and the larger number means the larger change; and the y-axis denoted the obtained average reward for 10 episodes, and the one standard deviation was also filled there. More results on MuJoCo environments are given in Figure 4 with four variations. Please see the Supplementary for detailed explanations of the designed variants. From Figure 7, we can observe that the rewards from PPO dramatically dropped from 500 (original environment) to around 300 starting from the first variation which was the smallest change. As increasing the variation, the performances kept dropping to a very low level and the standard deviation was surprisingly decreased as well. The shadow denotes the confidence of the algorithm on the prediction/performance, so the small shadows around the 4th and 5th environments denoted PPO failed on them and did not know its failure. On the contrary, our FWVPO was still very stable and obtained high rewards in variated environments. There was only a relatively small decrease from the 4th environment but a large variance was correctly

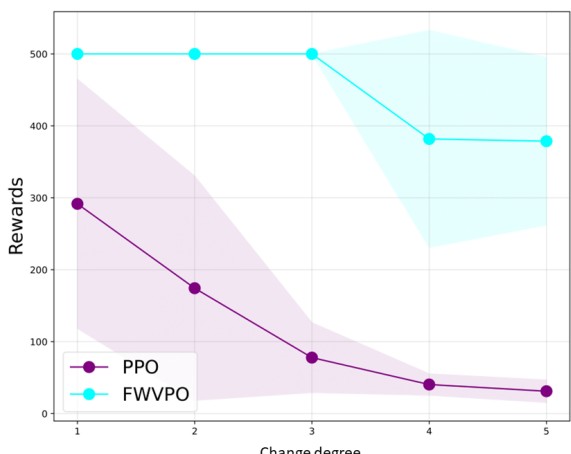

Figure 7: Evaluation on environment variations

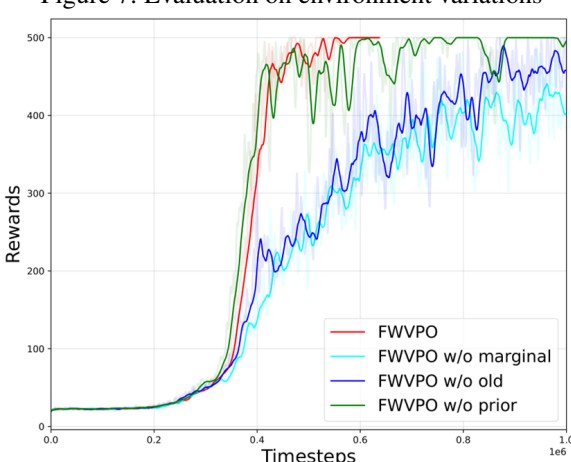

Figure 8: Contributions from three different terms

given to such decrease, which could support the following safe decision making. From Figure 4, we can also see that FWVPO consistently outperformed PPO on all four environments in terms of all variants. This fragility of PPO strongly motivates us to move to variational policy optimization.

## 5.5 ABLATION STUDY

We further studied the contributions from three terms in (12): distance with prior, distance with old posterior, and distance between marginal distributions. The ablation results are shown in Figure 8. At first, we can see that removing distance with the old posterior or distance between marginal distributions decreased the performance and the importance of distance with the old posterior was slightly higher than the marginal one. We also see that removing distance with prior did not decrease but slightly improved the performance. The reason is that we gave a non-informative prior in the experiments for simplicity. However, we could provide more meaningful prior practice by pretraining a prior using the randomly collected interactions or some other prior knowl-

edge of the policy or environment. More parameter analysis can be found in the Supplementary.

# 6 CONCLUSIONS AND FUTURE STUDIES

In this paper, we proposed a functional Wasserstein variational policy optimization (FWVPO) for reinforcement learning based on 1-Wasserstein distance instead of KL divergence and 2-Wasserstein distance. This new algorithm was empirically shown to have good capability in uncertainty modeling and generalization ability. We proved that FWVPO is a valid and tighter variational Bayesian objective and can also guarantee the monotonic expected reward improvement under certain conditions. Comparative experiments with several baselines on benchmark reinforcement learning tasks verified the proposed idea. In the future, we are going to further evaluate the proposed idea on the model-based RL where the functional BNNs would be used as the environment model [Lee et al., 2018] to increase the uncertainty modelling capability. Another interesting point is to investigate the possibility of properly expressing the 'probability density' for the function distribution.

### Acknowledgements

This work is supported by the Australian Research Council under the Discovery Early Career Researcher Award DE200100245 and Laureate Fellowships FL190100149.

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

# Functional Wasserstein Variational Policy Optimization
## (Supplementary Material)

**Junyu Xuan**[1]      **Mengjing Wu**[1]      **Zihe Liu**[1]      **Jie Lu**[1]

[1]Australian Artificial Intelligence Institute, University of Technology Sydney, Ultimo NSW 2007, Australia,

## A    SETTING FOR FIGURE 1 IN THE PAPER

We first define a Gaussian mixture model (GMM) as our target distribution,

$$p(x) = 0.1 * \mathcal{N}\left(x; \begin{bmatrix} 0 \\ 0 \end{bmatrix}, \begin{bmatrix} 2 & 0 \\ 0 & 2 \end{bmatrix}\right) + 0.2 * \mathcal{N}\left(x; \begin{bmatrix} 20 \\ 20 \end{bmatrix}, \begin{bmatrix} 3 & 0 \\ 0 & 3 \end{bmatrix}\right) + 0.7 * \mathcal{N}\left(x; \begin{bmatrix} -10 \\ 20 \end{bmatrix}, \begin{bmatrix} 1 & 0 \\ 0 & 1.5 \end{bmatrix}\right) \tag{13}$$

where three components are included with corresponding weights. The log-likelihood contour field is plotted in Figure 1. We then use a Gaussian distribution

$$q(x) = 0.1 * \mathcal{N}\left(x; \mu, \begin{bmatrix} 5 & 0 \\ 0 & 5 \end{bmatrix}\right) \tag{14}$$

to approximate the above-defined GMM distribution, where $\mu$ is the mean parameter that needs to be optimized. Finally, we use KL divergence ($\mathcal{KL}[q||p]$) and Wasserstein distance ($W_1[q, p]$) as the loss function to optimize $\mu$, respectively. The other hyperparameters are the same for all, like optimizer, steps, and learning rates.

The results are shown in Figure 1, where we set two different initializations (Figures 1a and 1b). We can see that

- KL divergence is sensitive to the initialization. For different initializations, there are two different results (Figures 1e and 1f) from KL divergence. In contrast, the results from Wasserstein distance (Figures 1c and 1d) are the same under different initializations.
- KL divergence tends to converge to the local optimal mode, which is consistent with the findings in [Neumann et al., 2011].
- Wasserstein could jump out of the local optimum and move close to the global optimal mode (which is the up-left corner one with the darkest color in Figure 1).

## B    NOTATION TABLE

In Table 1, we list the notations used in the paper and a description of their representation.

## C    SOME STANDARD CONCEPTS OF REINFORCEMENT LEARNING

The definitions of standard concepts follow the ones in TRPO [Schulman et al., 2015], including the state-action value function

$$Q_\pi(s, a) = \mathbb{E}_{s_{t+1}, a_{t+1}, \dots} \left[ \sum_{l=0}^{\infty} \gamma^l r(s_{t+l}) \right]$$

and state value function

$$V_\pi(s) = \mathbb{E}_{a_t, s_{t+1}, \dots} \left[ \sum_{l=0}^{\infty} \gamma^l r(s_{t+l}) \right]$$

Table 1: Notation table

| Notation | Description |
|----------|-------------|
| $p_0(f)$ | functional prior distribution |
| $q(f)$ | functional variational posterior distribution |
| $\theta$ | policy parameters, e.g., neural network weights for deterministic policy parameterization |
| $\vartheta$ | neural network weights' distribution parameters for BNN policy parameterization |
| $\vartheta^f$ | the neural network weights' distribution parameters for a BNN that induces a functional distribution $q(f)$ |
| $S$ | measurement set |

and the advantage function

$$A_\pi(s,a) = Q_\pi(s,a) - V_\pi(s).$$

# D  MORE DETAILS ABOUT DIFFERENT POLICY PARAMETERIZATIONS

- **Policy parameterized by a deterministic neural network.** The policy is traditionally parameterized by a deterministic deep neural network in which the final layer outputs parameters of an (action) distribution

$$\varpi(a|s; \theta) \tag{15}$$

where $\theta$ is neural network weights.

- **Policy parameterized by a BNN in weight space.** BNN was used to replace the deterministic deep neural network [Levine, 2018, Liu et al., 2017],

$$\pi(a|s) = \mathbb{E}_{p(\theta|\vartheta)}\left[\varpi(a|s; \theta)\right] \tag{16}$$

where $p(\theta|\vartheta)$ is the distribution of neural network weights, such as commonly used i.i.d. Gaussian distributions.

- **Policy parameterized by a BNN in function space.** We use BNN as the policy representation but work in the function space rather than the weight space,

$$\pi(a|s) = \mathbb{E}_{p(f)}\left[\varpi(a|f(s))\right] = \mathbb{E}_{p(\theta^f|\vartheta^f)}\left[\varpi(a|s; \theta^f)\right] \tag{17}$$

where $p(f)$ is a functional distribution induced by a parameterized BNN with $p(\theta^f|\vartheta^f)$. Since it is hard to represent a function for BNN, it is commonly adopted to use BNN weights to represent a function because there is a mapping between $f$ and $\theta^f$ and then a mapping between $p(f)$ and $p(\theta^f|\vartheta^f)$. In a nutshell, $p(f)$ can be simply understood as a BNN whose weights are with a distribution parameterized by $\vartheta^f$. It is important to note that although the calculation/evaluation of $\mathbb{E}_{p(f)}\left[\varpi(a|f(s))\right]$ and $\mathbb{E}_{p(\theta^f|\vartheta^f)}\left[\varpi(a|s; \theta^f)\right]$ looks like a policy parameterization in weight space, it is different from weight space parameterization in terms of regularization, such as KL divergence of (7) or Wasserstein distance of (8) in the paper. Similarly, the evaluation of $\mathbb{E}_{q(f)}\left[J(f)\right]$ of (7) and (8) in the paper also uses the inducing distribution,

$$\mathbb{E}_{q(f)}\left[J(f)\right] = \mathbb{E}_{q(\theta^f|\vartheta^f)}\left[J(\theta^f)\right] \tag{18}$$

where $J(\theta^f)$ can be $J^{\text{TRPO}}(\theta^f)$ or $J^{\text{PPO}}(\theta^f)$.

The pseudo-code of the whole procedure is briefly summarised in Algorithm 1.

**Algorithm 1** FWVPO

1: **Require** pool $\mathcal{M}$, and memory buffer $\mathcal{B}$,
2: Initialize a GP prior $G_0$ and a BNN parameterized by $\vartheta$
3: Initialize three Lipschitz functions $\phi_{\varphi'}, \phi_{\tilde{\varphi}}, \phi_{\hat{\varphi}}$ parameterized by $\varphi', \tilde{\varphi}, \hat{\varphi}$, respectively
4: **for** $t = 0, 1, \ldots$ **do**
5:      Draw a measurement set $S$ from the pool $\mathcal{M}$
6:      Combine $S = \{\mathcal{B}, S\}$
7:      Draw $N$ functions from $G_0$ on $S$, $\{f_i'(S)\}_{i=1:N}$
8:      Draw $N$ functions from $q_{\text{old}}(f)$ on $S$, $\{\tilde{f}_i(S)\}_{i=1:N}$
9:      Draw $N + M$ functions from $q_\vartheta(f)$ on $S$, $\{f_i(S)\}_{i=1:N}$ and $\{f_j(S)\}_{j=1:M}$
10:     Update $\varphi'$ by

$$\arg\max_{\varphi'} \left| \frac{1}{N} \sum_{i=1:N} \phi_{\varphi'}(f_i(S)) - \frac{1}{N} \sum_{i=1:N} \phi_{\varphi'}(f_i'(S)) \right|, \text{ s.t. } \|\phi_{\varphi'}\|_{L \leq 1}$$

11:     Update $\tilde{\varphi}$ by

$$\arg\max_{\tilde{\varphi}} \left| \frac{1}{N} \sum_{i=1:N} \phi_{\tilde{\varphi}}(f_i(S)) - \frac{1}{N} \sum_{i=1:N} \phi_{\tilde{\varphi}}(\tilde{f}_i(S)) \right|, \text{ s.t. } \|\phi_{\tilde{\varphi}}\|_{L \leq 1}$$

12:     Update $\hat{\varphi}$ by

$$\arg\max_{\hat{\varphi}} \left| \frac{1}{N} \sum_{i=1:N} \phi_{\hat{\varphi}}(f_i(S)) - \frac{1}{M} \sum_{j=1:M} \phi_{\hat{\varphi}}(f_j(S)) \right|, \text{ s.t. } \|\phi_{\hat{\varphi}}\|_{L \leq 1}$$

13:     Update $\vartheta$ by (11) in the paper
14: **end for**

# E   PROOF FOR THEOREM 1 IN THE PAPER

*Proof.* We prove the first inequality as below.

$$\mathbb{E}_{q(f^S)}\left[J(f^S)\right] - \frac{\rho}{2}\mathcal{W}^2[q(f^S)\|p_0(f^S)]$$

$$=\mathbb{E}_{q(f^S)}\left[J(f^S)\right] - \mathcal{KL}[q(f^S)\|p_0(f^S)] + \mathcal{KL}[q(f^S)\|p_0(f^S)] - \frac{\rho}{2}\mathcal{W}^2[q(f^S)\|p_0(f^S)]$$

$$= \log p(D) - \mathcal{KL}[q(f^S)\|p(f^S|D)] + \mathcal{KL}[q(f^S)\|p_0(f^S)] - \frac{\rho}{2}\mathcal{W}^2[q(f^S)\|p_0(f^S)]$$

$$= \log p(D) - \left( \mathcal{KL}[q(f^S)\|p(f^S|D)] - \mathcal{KL}[q(f^S)\|p_0(f^S)] + \frac{\rho}{2}\left( \max_{\|\phi\|_{L \leq 1}} \left\{ \mathbb{E}_{q(f^S)}[\phi(f^S)] - \mathbb{E}_{p_0(f^S)}[\phi(f^S)] \right\} \right)^2 \right) \tag{19}$$

Next, we only need to show the second term is positive. If $-\log p_0(f^S)$ is a special $\phi(f^S)$, we have

$$\mathcal{KL}[q(f^S)\|p(f|D)] - \mathcal{KL}[q(f^S)\|p_0(f^S)] + \frac{\rho}{2}\left( \max_{\|\phi\|_{L \leq 1}} \left\{ \mathbb{E}_{q(f^S)}[\phi(f^S)] - \mathbb{E}_{p_0(f^S)}[\phi(f^S)] \right\} \right)^2$$

$$\geq -\mathbb{E}_{q(f^S)}[\log p(f^S|D)] + \mathbb{E}_{q(f^S)}[\log p_0(f^S)] + \frac{\rho}{2}\left( \mathbb{E}_{q(f^S)}[\log p_0(f^S)] + \mathbb{E}_{p_0(f^S)}[\log p_0(f^S)] \right)^2$$

$$= \frac{\rho}{2}\left( \mathbb{E}_{q(f^S)}[\log p_0(f^S)] - \mathbb{E}_{p_0(f^S)}[\log p_0(f^S)] + \frac{1}{\rho} \right)^2 + \mathbb{E}_{p_0(f^S)}[\log p_0(f^S)] - \mathbb{E}_{q(f^S)}[\log p(f^S|D)] - \frac{1}{2\rho} \tag{20}$$

$$\geq \frac{\rho}{2}\left( \mathbb{E}_{q(f^S)}[\log p_0(f^S)] - \mathbb{E}_{p_0(f^S)}[\log p_0(f^S)] + \frac{1}{\rho} \right)^2 + H(q) - H(p_0) - \frac{1}{2\rho}$$

$$\geq 0$$

The second inequality is from the Talagrand inequality (It is from Definition 1 of the paper [Otto and Villani, 2000] which

further sources from Theorem 1.1 of the paper [Talagrand, 1996], and note that when the cost function is not with a square, the inequality should be Eq (1.2) of [Talagrand, 1996]): the probability measure $q$ satisfies a Talagrand inequality with constant $\rho > 0$ if for all probability measure $p$, absolutely continuous w.r.t. $q$, with finite moments of order 2,

$$W_1(p, q) \leq \sqrt{\frac{2\mathcal{KL}[p\|q]}{\rho}}. \tag{21}$$

Then, we can easily see that $\mathcal{L}^{\mathcal{W}} \geq \mathcal{L}^{\mathcal{KL}}$.

$\square$

## F  PROOF FOR THEOREM 2 IN THE PAPER

*Proof.* Following TRPO [Schulman et al., 2015], we define

$$\eta(\tilde{\pi}) = \eta(\pi) + \mathbb{E}_{\tau \sim \tilde{\pi}} \left[ \sum_{t=0}^{\infty} \gamma^t A_\pi(s_t, a_t) \right]$$

$$L_\pi(\tilde{\pi}) = \eta(\pi) + \mathbb{E}_{\tau \sim \pi} \left[ \sum_{t=0}^{\infty} \gamma^t A_\pi(s_t, a_t) \right] \tag{22}$$

and we know that

**Theorem 3** (Theorem 2.1 in [Chae and Walker, 2020]).

$$(TV(\bar{\pi}, \pi))^{(\beta+1)/\beta} \leq c \left( \|\bar{\pi}\|_{H_1^\beta} + \|\pi\|_{H_1^\beta} \right)^{1/\beta} \mathcal{W}(\bar{\pi}, \pi)$$

where $c > 0$ is a constant, $\beta \in \mathbb{N}$ is independent with two distributions, and $\|f\|_{H_1^\beta} = \|f\|_1 + \|\nabla^\beta f\|_1$.

According to the above theorem, when the $\beta \to \infty$, we have $TV(\bar{\pi}, \pi) \leq c\mathcal{W}(\bar{\pi}, \pi)$. Then,

$$\eta(\pi_{new}) \geq L_{\pi_{old}}(\pi_{new}) - \frac{4\gamma\epsilon}{(1-\gamma)^2} \left( TV^{\max}(\pi_{old}\|\pi_{new}) \right)^2$$

$$\geq L_{\pi_{old}}(\pi_{new}) - \frac{4\gamma\epsilon}{(1-\gamma)^2} \lambda(1) \left( \mathcal{W}^{\max}(\pi_{old}\|\pi_{new}) \right)^2 \tag{23}$$

$$= L_{\pi_{old}}(\pi_{new}) - \frac{4\gamma\epsilon}{(1-\gamma)^2} \lambda(1) \left( \sup_\phi \left( \mathbb{E}_{f \sim \tilde{p}(f)} \mathbb{E}_{a \sim \varpi(a;\theta^f)} [\phi(a)] - \mathbb{E}_{f \sim p(f)} \mathbb{E}_{a \sim \varpi(a;\theta^f)} [\phi(a)] \right) \right)^2$$

where $\epsilon = \max_{s,a} |A_\pi(s, a)|$, $\lambda(1) = c^2$ where 1 denotes $\beta = 1$; and we ignore the coefficient without loss of generality because it can be easily adjusted to match the coefficient. Since $\phi$ can be any Lipschitz function, we next assume that the optimal one to maximize $\mathbb{E}_{f \sim \tilde{p}(f)} \mathbb{E}_{a \sim \varpi(a;\theta^f)} [\phi(a)] - \mathbb{E}_{f \sim p(f)} \mathbb{E}_{a \sim \varpi(a;\theta^f)} [\phi(a)]$ is $\phi^*$ and $\mathbb{E}_{a \sim \varpi(a;\theta^f)} [\phi^*(a)]$ is also Lipschitz function of $f$. Then, we have

$$L_{\pi_{old}}(\pi_{new}) - \frac{4\gamma\epsilon}{(1-\gamma)^2} \lambda(1) \left( \sup_\phi \left( \mathbb{E}_{f \sim \tilde{p}(f)} \mathbb{E}_{a \sim \varpi(a;\theta^f)} [\phi(a)] - \mathbb{E}_{f \sim p(f)} \mathbb{E}_{a \sim \varpi(a;\theta^f)} [\phi(a)] \right) \right)^2$$

$$\geq L_{\pi_{old}}(\pi_{new}) - \frac{4\gamma\epsilon}{(1-\gamma)^2} \lambda(1) \left( \mathcal{W}^{\max}[\tilde{p}(f), p(f)] \right)^2 \tag{24}$$

which proves the Theorem with only a difference in the coefficient of the Wasserstein distance term. We can easily absorb $4\gamma\epsilon$ into $\phi$ function definition and then obtain the same results.

Note that both $p(f)$ and $\tilde{p}(f)$ do not depend on $s$, but $\mathcal{W}$ depends on it because $\phi(s) = \mathbb{E}_{\varpi(a|s;\theta^f)}[A_\varpi(s, a)]$ depends on $s$. The 'max' in (24) is not a big problem because we can remove it. The reason is that we assumed all $\phi(s)$ are Lipschitz functions, so $\mathcal{W}$ is the supremum of distances defined by all candidate $\phi(s_t)$ for all $t$. The reason why we kept it here is to ease the comparison with TRPO.

Table 2: Hyperparameters for CartPole and Acrobot

| Name | Value |
|---|---|
| max time steps in one episode | 500 |
| update policy frequency | 2,000 (unless otherwise specified) |
| number of epochs for policy update | 80 |
| number of steps for Lipschitz function maximization | 10 |
| clip parameter for PPO | 0.2 |
| discount factor $\gamma$ | 0.99 |
| activation function | Tanh |
| learning rate for actor network | 0.0003 |
| learning rate for critic network | 0.001 |
| learning rate for Lipschitz function | 0.01 |
| random seed | 12 (unless otherwise specified) |

Table 3: Hyperparameters for MuJoCo experiments

| Name | Value |
|---|---|
| max time steps in one episode | 2048 |
| update policy frequency | 5 episodes |
| number of epochs for policy update | 10 |
| number of steps for Lipschitz function maximization | 10 |
| clip parameter for PPO | 0.2 |
| discount factor $\gamma$ | 0.99 |
| activation function | Tanh |
| learning rate for actor-network | 0.0003 |
| learning rate for critic-network | 0.0003 |
| learning rate for Lipschitz function | 0.01 |
| random seed | 12 |

For the second half of the Theorem (the relationship between KL divergence), we use Talagrand inequality [Otto and Villani, 2000] again: the probability measure $q$ satisfies a Talagrand inequality with constant $\rho$ if for all probability measure $p$, absolutely continuous w.r.t. $q$, with finite moments of order 2,

$$W_1(p, q) \leq \sqrt{\frac{2\mathcal{KL}[p\|q]}{\rho}}. \tag{25}$$

Then, we can easily see that

$$
\begin{aligned}
\eta_{KL} &= L_{\pi_{\text{old}}}(\pi_{\text{new}}) - \frac{4\gamma\epsilon}{(1-\gamma)^2}\mathcal{KL}[\pi_{\text{old}}\|\pi_{\text{new}}] \\
&\leq L_{\pi_{\text{old}}}(\pi_{\text{new}}) - \frac{4\gamma\epsilon}{(1-\gamma)^2}\frac{\rho}{2}\left(\mathcal{W}[p_{\text{old}}(f)\|p_{\text{new}}(f)]\right)^2 \\
&= \eta_{\mathcal{W}}.
\end{aligned} \tag{26}
$$

# G  MORE DETAILS FOR THE EXPERIMENTS IN THE PAPER

## G.1  SETUP DETAILS

The evaluation environments were from Gym[1]. The PPO was used as the base model [Barhate, 2021], including an actor-network and a critic network. The network architecture for the actor was Linear(input, 64)-Identity(64)-Tanh-Linear(64,

---

[1]https://www.gymlibrary.ml/

64)-Identity(64)-Tanh-Linear(64, output) and a Softmax was added for discrete actions; the architecture for the critic network was Linear(input, 64)-Tanh-Linear(64, 64)-Tanh-Linear(64, 1). All algorithms shared exactly the same critic network. The basic actor network was also the same but BNN-based algorithms were assigned prior to the network parameters. The used hyperparameters are given in Table 2. Apart from the basic control environments, we also tested our proposed algorithm on MuJoCo benchmarks [2], including Hopper and Humanoid. Here, we used two-layer BNN in FWVPO and more hyperparameters are given in Table 3.

The 2-Wasserstein distance (also known as Fréchet distance [Dowson and Landau, 1982]) between two Gaussian distributions used by **BNN-W-PPO** was evaluated as

$$\mathcal{W}_2^2 = |\mu_X - \mu_Y|^2 + tr(\Sigma_X + \Sigma_Y - 2(\Sigma_X \Sigma_Y)^{1/2})$$

and the KL divergence between function samples used by **fBNN-KL-PPO** was evaluated using grid KL in [Ma and Hernández-Lobato, 2021] where a geometric distribution was firstly used to sample a measure set size and then a number of observations were uniformly sampled from the buffer and then the spectral stein gradient estimator [Shi et al., 2018] was used to estimate the KL divergence between marginal distributions on measurement set. FWVPO collected a set of states in memory before training and used it as the random global measurement set, where the collection was implemented using a random policy to interact with the environment. The Wasserstein distance optimization is based on the code from [Tran et al., 2022][3].

## G.2 SETUP DETAILS FOR NOISY OBSERVATIONS

To simulate noisy environments in the paper, random noise was added to the observed state and then fed to RL agents for training at each time step. The values were selected to impact the performance of the base model (PPO) significantly.

- For **Acrobot**, we used multivariate normal distribution for noise generation:

$$\mathcal{N}\left(\begin{bmatrix} 0 \\ 0 \\ 0 \\ 0 \\ 0 \\ 0 \end{bmatrix}, \begin{bmatrix} 0.5 & 0 & 0 & 0 & 0 & 0 \\ 0 & 0.5 & 0 & 0 & 0 & 0 \\ 0 & 0 & 0.5 & 0 & 0 & 0 \\ 0 & 0 & 0 & 0.5 & 0 & 0 \\ 0 & 0 & 0 & 0 & 10 & 0 \\ 0 & 0 & 0 & 0 & 0 & 15 \end{bmatrix}\right).$$

- For **CartPole**, we used $\mathcal{N}\left(\begin{bmatrix} 0 \\ 0 \\ 0 \\ 0 \end{bmatrix}, \begin{bmatrix} 1 & 0 & 0 & 0 \\ 0 & 1 & 0 & 0 \\ 0 & 0 & 0.1 & 0 \\ 0 & 0 & 0 & 1 \end{bmatrix}\right).$

- For **Hopper**, **Walker2d** and **Humanoid**, we used $\mathcal{N}(\mathbf{0}, 0.1 * \mathbf{I})$, where $\mathbf{I}$ is the identify matrix.
- For **Halfcheetah**, we used $\mathcal{N}(\mathbf{0}, 2 * \mathbf{I})$, where $\mathbf{I}$ is the identify matrix.

## G.3 SETUP DETAILS FOR ENVIRONMENT VARIATIONS

- For **CartPole**[4], we revised its some parameters to obtain the changed environments. We firstly changed its transition by revising its one line of *step()* from

    temp = (force + self.polemass_length * theta_dot**2 * sintheta) / self.total_mass

    to

    temp = (force + self.polemass_length * theta_dot**4 * sintheta) / self.total_mass

    and we also changed the *self.gravity = 9.8* to *self.gravity = 9.8 + x* where **x** was set as 5, 10, 15, 20, and 25. Such variates are expected to change the underlying dynamics.

---

[2]https://www.gymlibrary.dev/environments/mujoco/index.html
[3]https://github.com/tranbahien/you-need-a-good-prior
[4]https://github.com/openai/gym/blob/master/gym/envs/classic_control/cartpole.py

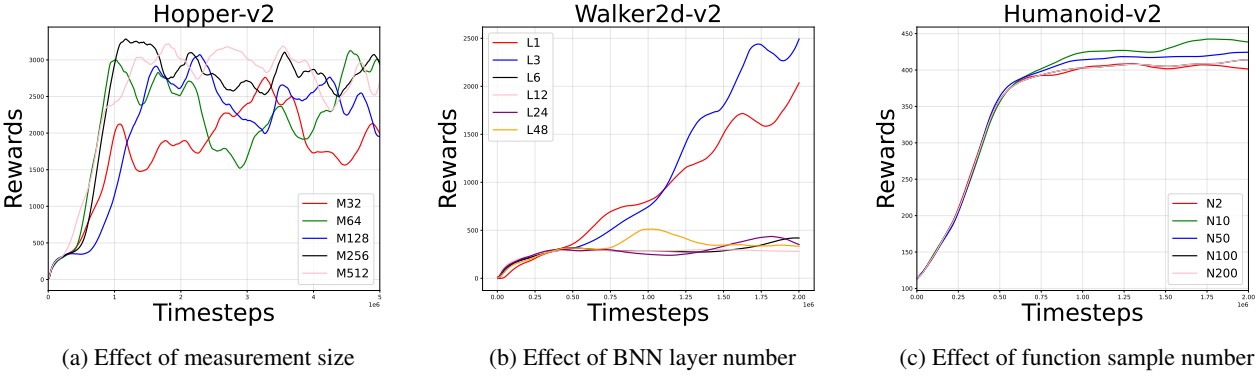

Figure 9: Parameter sensitivity analysis

- For **Hopper**, we changed its reward calculation by revising its one line of *step()* from

    reward -= 1e-3 * np.square(a).sum()

  to

    reward -= x * np.square(a).sum()

  where x was set as 1e-3, 1e-2, 0.1 and 0;

- For **Humanoid**, we firstly changed its transition by revising its one line of *step()* from

    lin_vel_cost = 1.25 * (pos_after - pos_before) / self.dt

  to

    lin_vel_cost = x * (pos_after - pos_before) / self.dt

  and also changed

    quad_ctrl_cost = 0.1 * np.square(data.ctrl).sum()

  to

    quad_ctrl_cost = y * np.square(data.ctrl).sum()

  where x was set as 3.25, 5.25, 7.25 and 9.25 and y was set as 0.01, 0.001, 0.001 and 0.001.

- For **Walker2d**, we firstly revised its one line of *step()* from

    alive_bonus = 1.0

  to

    alive_bonus = x

  and also changed

    reward -= 1e-3 * np.square(a).sum()

  to

    reward -= y * np.square(a).sum()

  where $x$ was set as 1.0, 2.0, 3.0 and 4.0 and $y$ was set as $1e-3$, $1e-2$, $1e-1$ and 0.05.

- For **Halfcheetah**, we revised its one line of *step()* from

    reward_ctrl = -0.1 * np.square(action).sum()

  to

    reward_ctrl = -x * np.square(action).sum()

  where x was set as 0.1, 0.15, 0.01, and 0.2.

# H  PARAMETER SENSITIVITY ANALYSIS

We studied the contributions from three hyperparameters: measurement size, BNN layer number and function sample number. The effect from measurement size (Line5 of Algorithm 1) is shown in Figure 9a, where we observed that increasing the number of measurement sizes could generally improve the performance. For example, 256 and 512 were better than 32 and 64, while there is not much difference between 256 and 512. We used 64 as the default for previous experiments. The effect from the number of BNN layers is shown in Figure 9b, where we observed that more BNN layers would take more steps to get converged so the performance of one and three layers was better than the other four options within 2e6 steps. Among one and three layers, the three-layer one was better. We used 3 as default for previous experiments. The effect from the number of function samples (Lines 7 and 8 of Algorithm 1) is shown in Figure 9c, where we observed that 10 is the best among all options and we used it as the default for the previous experiments.