# OpenReview forum: "Functional Wasserstein Variational Policy Optimization"
_auai.org/UAI/2024/Conference — UAI 2024 poster_

### Official Review · Reviewer_TqGX · 2024-03-21

**Q2-1 Originality-Novelty:** 2
**Q2-2 Correctness-Technical Quality:** 1
**Q2-5 Clarity Of Writing:** 3

**Q1 Summary And Contributions:**

They proposed a new functional Wasserstein variational inference based on 1-Wasserstein distance rather than KL divergence and showed that their algorithm FWVPO has the monotonic improvement compared to KL-divergence.

**Q2-3 Extent To Which Claims Are Supported By Evidence:**

1: Poor: the authors fail to convincingly backup their main claims (e.g., if the experimental evaluation is flawed, proofs are lacking or invalid, references are missing, assumptions are not realistic, not specified, or not motivated).

**Q2-4 Reproducibility:**

3: Good: key resources (e.g. proofs, code, data) are available and key details (e.g. proofs, experimental setup) are sufficiently well-described for competent researchers to confidently reproduce the main results.

**Q3 Main Strengths:**

Using KL-based functional variational inference might pose challenges due to the non-monotonic improvement at each step, potentially leading to instability. To mitigate this issue, they replaced KL-divergence with the square of the l1-Wasserstein distance in the regularization. Additionally, they include a promising experimental section for FWVPO, although I have not reviewed their code.

**Q4 Main Weakness:**

The theoretical parts of this paper are written poorly. For instance, policy function $f$ is not defined in Section 3. In Theorem 1, did you consider a special choice of reward function in $J(f)$ in the definition of  $\mathcal{L}^{W}$? What is $p(D)$ in Theorem 1?
1) Why did you impose constraints on the underlying optimization of $\mathcal{L}^{W}$ in (8)? You initially defined $\mathcal{L}^{W}$ without any constraints in (5).
2) The proofs of Theorems 1 and 2 are inadequately presented. In the proof of Theorem 1, I find it difficult to understand the derivation of Equation (19). Could you provide more detail on this step?
3) About the second inequality that you mentioned comes from Talagrand inequality, how did you verify that distribution $q$ satisfies T1 inequality? I did not find any T1 inequality assumption about the distribution $q$.
4) In the proof of Theorem 2, I'm unsure about how you derived the first inequality after Theorem 3, and when did you define $\epsilon$? What is $\lambda(1)$ in the second inequality after Theorem 3? Do you need any assumption to make sure that the term ||\tilde{\pi}||_{H_{1}^{\beta}}+||\pi||_{H_{1}^{\beta}} is bounded? Again did you assume that $q$ satisfies T1 inequality for Theorem 2?

**Q5 Detailed Comments To The Authors:**

They are provided in Main weakness.

**Q9 Complying With Reviewing Instructions:**

Yes

---

> ### Author Rebuttal · Authors · 2024-04-05
>
> ## We thank you for the comments and give the corresponding responses as follows. If you are satisfied with these responses, please consider raising your score accordingly.
>
> > policy function $f$ is not defined in Section 3. In Theorem 1, did you consider a special choice of reward function in $J(f)$ in the definition of $\mathcal{L}^W$? What is $p(D)$ in Theorem 1?
>
> **Reply**: $f$ is the policy function which is shown under Eq (4), and a policy is normally a mapping from state to action in RL. In theorem 1, we do not consider a specific choice of $J(f)$ and any surrogate term of TRPO or PPO should be fine, and we used the PPO one in our implementation.
> We define $p(D|f) = \exp{(J(D, f))}$ where $D$ denotes the collected episodes/data and we include it here to make it more clear, and $p(D) =  \int_f p(D|f) \rm{d} f = \int_f \exp{(J(D, f))} \rm{d} f$. The above definitions are similar to the MEPO but with a function rather than a vector as the latent variable. Note that the reviewer may be concerned that the `likelihood' here is not a well-specific probability. In fact, it can be any cost function of latent variable and observation in the generalized variational inference framework and more details about this framework can be found in paper [1].
>
>
> > Why did you impose constraints on the underlying optimization of $\mathcal{L}^W$ in (8)? You initially defined $\mathcal{L}^W$ without any constraints in (5).
>
> **Reply**: Instead of directly giving the final objective function, we progressively introduce our idea from (4) to (5) and final (12) to ease the understanding. (4) is used to introduce the idea of replacing the original weight-space KL divergence with a function-space KL divergence, while (5) is used to introduce the idea of replacing the function-space KL divergence with a function-space Wasserstein distance. Then, (8) together with the constraint is introduced to ensure that it is a valid variational objective function.
>
>
> > In the proof of Theorem 1, I find it difficult to understand the derivation of Equation (19). Could you provide more detail on this step?
>
> **Reply**: Firstly, we define $p(D|f) = \exp{(J(D, f))}$ where $D$ denotes the observed episodes and we include it here to make it clearer and $J(D, f)$ could be any surrogate term, and $p(D) =  \int_f p(D|f) \rm{d} f = \int_f \exp{(J(D, f))} \rm{d} f$. The core part of (19) just follows the classical evidence lower bound derivation, and we show the details as follows:
>
> $$E_q [ J(D, f) ] - \mathcal{KL}[q(f) \| p_0(f)]$$
>
> $$=E_q[ \log p(D|f)  ] - \mathcal{KL}[q(f) \| p_0(f)] $$
>
> $$=E_q[ \log p(D|f) ] - E_q[ \log q(f) ]  + E_q [ \log p_0(f) ]$$
>
> $$=E_q[ \log p(D|f)  ] + E_q[ \log p_0(f) ] - E_q [ \log q(f) ] $$
>
> $$=E_q[ \log p(D) ]  + E_q[ \log p(f|D) ] - E_q [ \log q(f) ] $$
>
> $$=\log p(D) - E_q[ \log \frac{q(f)}{p(f |D)} ] $$
>
> $$=\log p(D) - \mathcal{KL}[q(f) \| p(f |D)] $$
>
> We will add these details in the updated version.
>
>
> > how did you verify that distribution satisfies T1 inequality? I did not find any T1 inequality assumption about the distribution $q$. Again did you assume that $q$ satisfies T1 inequality for Theorem 2?
>
> **Reply**: It is from Definition 1 of the paper [2] which further sources from Theorem 1.1 of the paper [3] (Note that when the cost function is not with a square power, the inequality should be Eq (1.2) of [3]). The condition of this inequality is that the probability measure $p$ is absolutely continuous w.r.t. $q$ with finite moments of order 2. Such a condition is mainly used to ensure the KL divergence is well-defined, so it is only a mild condition for general situations. We will add such a condition to the Theorem 1 of the paper.
>
> > In the proof of Theorem 2, I'm unsure about how you derived the first inequality after Theorem 3, and when did you define $\epsilon$? What is $\lambda(1)$ in the second inequality after Theorem 3? Do you need any assumption to make sure that the term $||\tilde{\pi}||{H{1}^{\beta}}+||\pi||{H{1}^{\beta}}$ is bounded?
>
> **Reply**: It is our fault to miss these definitions and details. Most concepts of this paper follow the ones from TRPO, including $\epsilon = \max_{s,a}|A_{\pi}(s,a)|$. The $\lambda(1) = c^2$ where $1$ denotes $\beta=1$. The condition to ensure bounded term is $0 < ||\tilde{\pi}||_1 < \infty, 0 < ||\pi||_1 < \infty$. We will include these details in the new version of the paper.
>
> &nbsp;
> &nbsp;
>
> ## References
>
> [1]. Knoblauch, Jeremias, et al. "An optimization-centric view on Bayes' rule: Reviewing and generalizing variational inference." JMLR, no. 132 (2022): 1-109.
>
> [2]. Otto, Felix, and Cédric Villani. "Generalization of an inequality by Talagrand and links with the logarithmic Sobolev inequality." Journal of Functional Analysis 173, no. 2 (2000): 361-400.
>
> [3]. Talagrand, Michel. "Transportation cost for Gaussian and other product measures." Geometric \& Functional Analysis GAFA 6, no. 3 (1996): 587-600.

---

### Official Review · Reviewer_2Qe5 · 2024-03-22

**Q2-1 Originality-Novelty:** 2
**Q2-2 Correctness-Technical Quality:** 3
**Q2-5 Clarity Of Writing:** 2

**Q1 Summary And Contributions:**

This work replaces the KL divergence in the TRPO method with the Wasserstein-1 distance. The derived algorithm has the advantage that one policy is not absolutely continuous with respect to another. Also, this paper shows that the new objective function provides a new variational Bayesian perspective. Lastly, this paper shows that the new objective function follows the same monotonic improvement as TRPO.

**Q2-3 Extent To Which Claims Are Supported By Evidence:**

3: Good: the main claims are supported by convincing evidence (in the form of adequate experimental evaluation, proofs, (pseudo-)code, references, assumptions).

**Q2-4 Reproducibility:**

2: Fair: key resources (e.g. proofs, code, data) are unavailable but key details (e.g. proof sketches, experimental setup) are sufficiently well-described for an expert to confidently reproduce the main results.

**Q3 Main Strengths:**

This work provides a new perspective in understanding the proposed objective function.

**Q4 Main Weakness:**

This method is not new. Replacing the KL-divergence with another distance (especially the Wasserstein distance) has been studied before, e.g. Eq.(8) in [Richemond2017], Eq.(6) in [Zhang2018], and Problem (P) in [Terpin2022].

The policy improvement lemma also has been proved before. See Proposition 4 in [Terpin2022].

Missing details. I am not clear how you optimize your proposed objective function.

*References*:
* [Richemond2017] Richemond, Pierre H., and Brendan Maginnis. "On Wasserstein reinforcement learning and the Fokker-Planck equation." arXiv preprint arXiv:1712.07185 (2017).
* [Zhang2018] Zhang, Ruiyi, et al. "Policy optimization as wasserstein gradient flows." International Conference on machine learning. PMLR, 2018.
* [Terpin2022] Terpin, Antonio, et al. "Trust region policy optimization with optimal transport discrepancies: Duality and algorithm for continuous actions." Advances in Neural Information Processing Systems 35 (2022): 19786-19797.

**Q5 Detailed Comments To The Authors:**

1. The motivation of replacing KL-divergence with W1 distance is not convincing. If all policies are approximated using softmax, then they will be absolutely continous w.r.t each other.
2. This work has not addressed the obvious challenges in making the proposed replacement. For example, evaluating the objective function (5) requires to solve an optimal transport problem when calculating the W1 distance, which is usually hard. How do you solve that?
3. When evaluating the gradient of (12) to maximize the objective function, what is the gradient with respect to $f$ or $q$? Do you just parameterize everything?
4. Theorem 2 only guarantees the monotonic improvement. Does it ensure that the policy will converge to the optimal policy?

**Q9 Complying With Reviewing Instructions:**

Yes

---

> ### Author Rebuttal · Authors · 2024-04-05
>
> ## We thank you for the comments and give the corresponding responses as follows. If you are satisfied with these responses, please consider raising your score accordingly.
>
> > This method is not new. Replacing the KL-divergence with another distance (especially the Wasserstein distance) has been studied before, ....
>
>
> **Reply**: As discussed in the second paragraph of Section 4 (related work) where the given second reference is also included, we know there are already works trying to use Wasserstein distance to replace KL divergence for policy optimization. The difference is that our method is from a function-space view, especially the Wasserstein distance used in our method is used for function distributions rather than normal deterministic neural networks. All existing works including the ones discussed in the second paragraph of Section 4 and the ones given by the reviewer still use the deterministic neural network as policy parameterization, and hence their policy improvement proofs are different from ours if you compare the proof procedures.
>
>
> > The motivation of replacing KL-divergence with W1 distance is not convincing. If all policies are approximated using softmax, then they will be absolutely continous w.r.t each other.
>
> **Reply**: Other works also identified the problem of KL divergence that may be ill-defined in some situations, including the reference [6] given by the reviewer. Even under the softmax approximation assumed by the reviewer, KL divergence may also be ill-defined. We can give a simple example here: Suppose there is an agent that has four actions (up, down, left, right) in a 2D environment. In an episode, we collect several interactions from this agent in this environment where all actions are either up or down. Then, the policy trained on this data would output an action distribution (after softmax), like $[0.700, 0.299, 0.000, 0.000]$. In a new episode, we collect another set of interactions where all actions are either left or right. Then, the new policy would output an action distribution (after softmax), like $[0.000, 0.000, 0.499, 0.500]$. The KL divergence between these two (discrete) distributions would be extremely large (near infinity). This simple example shows that even though the potential spaces of two distributions are the same, the policies trained on two datasets may not share the same \textit{empirical spaces} and then the KL divergence on such non-over lapping \textit{empirical spaces} would be ill-defined, and the possibility is even higher for large state and action spaces and strong exploration in RL.
>
>
> > evaluating the objective function (5) requires solving an optimal transport problem when calculating the W1 distance, which is usually hard. How do you solve that?
>
> **Reply**: The calculation and optimization of W1 distance is within the optimization of (12), and the procedure is summarised in Algorithm 1 on Page 14. We follow [3] to use the differentiable gradient norm penalty method [2] to ensure the Lipschitz constraint of $\phi$. Such a method works well empirically. We notice that there are some advanced methods proposed in the literature for this problem as well [4, 5]. Those methods could further improve the efficiency of our algorithms.
>
>
> > When evaluating the gradient of (12) to maximize the objective function, what is the gradient with respect to $q$ or $f$? Do you just parameterize everything?
>
> **Reply**: $q$ is a distribution of functions induced by a BNN, so we only need to optimize $q$ (the parameter of BNN) during the training. $f$ here denotes the latent function defined by BNN, so we do not optimize it.
>
> > Theorem 2 only guarantees the monotonic improvement. Does it ensure that the policy will converge to the optimal policy?
>
> **Reply**: No, theorem 2 is similar to the one given in TRPO and other TRPO-based or PPO-based algorithms (including the given reference [6]) and can only guarantee theoretical improvement but no guarantee of reaching global optimal policy, which normally requires further assumptions on the reward function, action-value function class, and regularity of stationary distribution [1].
>
> &nbsp;
> &nbsp;
>
> ## References
>
> [1] Liu, Boyi, et al. "Neural proximal/trust region policy optimization attains globally optimal policy." arXiv preprint arXiv:1906.10306 (2019).
>
> [2] Gulrajani, Ishaan, et al. "Improved training of wasserstein gans." NIPS 30 (2017).
>
> [3] Tran, Ba-Hien, et al. "All you need is a good functional prior for Bayesian deep learning." JMLR 23, no. 74 (2022): 1-56.
>
> [4] Fazlyab, Mahyar, et al. "Efficient and accurate estimation of lipschitz constants for deep neural networks." NIPS 32 (2019).
>
> [5] Shi, Zhouxing, et al. "Efficiently computing local lipschitz constants of neural networks via bound propagation." NIPS 35 (2022): 2350-2364.
>
> [6] Terpin, Antonio, et al. "Trust region policy optimization with optimal transport discrepancies: Duality and algorithm for continuous actions." NIPS 35 (2022): 19786-19797.

---

### Official Review · Reviewer_rYXD · 2024-03-23

**Q2-1 Originality-Novelty:** 2
**Q2-2 Correctness-Technical Quality:** 3
**Q2-5 Clarity Of Writing:** 3

**Q1 Summary And Contributions:**

This paper explores functional variational policy optimization using the 1-Wasserstein distance instead of KL-divergence, to avoid some of the undesirable properties of KL. The authors have provided a practical implementation and demonstrate its effectiveness, particularly under noisy observations, demonstrating the benefits of approaching policy optimization from a Bayesian perspective.

**Q2-3 Extent To Which Claims Are Supported By Evidence:**

2: Fair: the main claims are somewhat supported by evidence (but the experimental evaluation may be weak, or does not match entirely with the claims, important baselines may be missing, proofs contain important ideas but lack rigor, algorithmic details are only discussed superficially, references are imprecise, assumptions are not sufficiently motivated or explicated, etc.).

**Q2-4 Reproducibility:**

2: Fair: key resources (e.g. proofs, code, data) are unavailable but key details (e.g. proof sketches, experimental setup) are sufficiently well-described for an expert to confidently reproduce the main results.

**Q3 Main Strengths:**

- The paper is well-written and generally easy to follow.

- To the best of my knowledge, functional variational policy optimization using 1-Wasserstein is new.

- The experiments with noisy observations demonstrate its effectiveness in modeling under uncertainty.

- Theoretical analysis is provided, for example, guarantee of monotonic improvement.

**Q4 Main Weakness:**

- The choice of baselines is somewhat limited. A comparison with methods   uncertainty-aware policy optimization methods (such as [1] or any reasonble alternatives) could help the audience better appreciate the advantages of fBNN-W-PPO.


[1] Lee, Gilwoo, et al. "Bayesian policy optimization for model uncertainty." arXiv preprint arXiv:1810.01014 (2018).

**Q5 Detailed Comments To The Authors:**

- While the experiments with noise illustrate the strengths of fBNN-W-PPO, it would be beneficial if the authors could further elaborate on and highlight BNN's advantages in uncertainty modeling, especially for readers less familiar with BNN literature.

**Q9 Complying With Reviewing Instructions:**

Yes

---

> ### Author Rebuttal · Authors · 2024-04-05
>
> ## Thanks for the comments and suggestions! If you are satisfied with our response and also the ones to other reviewers, please consider raising your score.
>
>
> > Further elaborate on and highlight BNN's advantage in uncertainty modelling, like the paper "Bayesian policy optimization for model uncertainty"
>
>
> **Reply**: Firstly, we evaluated and demonstrated the BNN's advantage in uncertainty modelling in this paper through two following tasks: noisy observations and environment generalization. The better uncertainty modelling would lead to better performance in these two tasks. Then, the given paper is in a different setting from ours. In the given paper, a belief distribution on transition probability and value function (or their parameters) is maintained and updated, which can be seen as a kind of combined approach of model-based RL and model-free RL. However, our method is a pure model-free RL algorithm.
>
> We thank the reviewer's suggestion to apply our idea to this interesting setting and we will consider it as our future work.

---

### Official Review · Reviewer_rj1K · 2024-03-25

**Q2-1 Originality-Novelty:** 3
**Q2-2 Correctness-Technical Quality:** 3
**Q2-5 Clarity Of Writing:** 2

**Q1 Summary And Contributions:**

This paper proposes a policy-gradient reinforcement learning (RL) algorithm based on regularisation via Wasserstein distances in function space. The approach is based on TRPO [Schulman et al., 2015] and PPO [Schulman et al., 2017], swapping the regularisation constraints by Wasserstein distance terms. Theoretical analyses show monotonic improvement guarantees similar to TRPO. Experiments evaluate the PPO variants of the algorithm on a set of benchmark functions for RL against a few baselines.

**Q2-3 Extent To Which Claims Are Supported By Evidence:**

3: Good: the main claims are supported by convincing evidence (in the form of adequate experimental evaluation, proofs, (pseudo-)code, references, assumptions).

**Q2-4 Reproducibility:**

3: Good: key resources (e.g. proofs, code, data) are available and key details (e.g. proofs, experimental setup) are sufficiently well-described for competent researchers to confidently reproduce the main results.

**Q3 Main Strengths:**

* Experiments show apparently significant performance improvements against baselines.
* Application of Wasserstein distance to RL approaches based on KL divergence allows for more robustness and algorithmic stability.
* Theoretical analysis and guarantees for the algorithms are provided.

**Q4 Main Weakness:**

* Some technical details in the methodology's derivation are unclear (see detailed comments below).
* Quality of writing could be improved, as currently there are quite a few typos and grammar issues, which make the paper hard to read.
* The proposed algorithm has quite a few moving parts and hyper-parameters, and it's unclear how to select all of them.
* It's not possible to assess the statistical significance of the improvements observed in the experimental results, since there seems to be no mention of whether multiple runs were used, and if so, how many.

**Q5 Detailed Comments To The Authors:**

### Background
* What is $s$ taken to be in Eq. 2? The KL term in the TRPO objective is missing an expectation over the states or a maximum, as in the original paper.

### Theorem 1
* Theorem 1 is missing assumptions used in the proof, such as the requirement for $q$ to satisfy Talagrand's inequality (unless that's implicitly satisfied by some other condition) and that $\log p_0(f)$ needs to be a **1**-Lipschitz function (though it might be possible to guarantee it by some sort of scaling).
* There's no discussion on the feasibility of satisfying the assumptions of Theorem 1 in a practical scenario. In particular, the Lipschitz condition on $\log p_0(f)$ seems quite strong, since that'd not be satisfied by some commonly used priors, such as Gaussians (in the finite-dimensional setting).
* The "evidence" term $\log p(D)$ in Theorem 1 is not explicitly defined in the main text, and there's no comment about its meaning, since we don't have a proper likelihood function (i.e., based on a conditional probability distribution) in this formulation.

### Final formulation (Eq. 12)
- It's confusing how the authors got the final formulation in Eq. 12, since up to that point only a single Wasserstein-based constraint was used in the objective function formulations, and now 3 of them appeared. I somewhat understand the 3 regularisation terms, but I believe that part of the text needs a smoother transition to and perhaps some more condensed justification for including the 3 constraints.
* Are the theoretical guarantees (monotonic improvement) still valid when the 3 regularisation terms are combined?
* How are the hyper-parameters ($\alpha_{1,..., 3}$ and $\rho$) selected for the experiments? Is there any ideal hyper-parameter selection procedure for other practical settings?

### Related work
* There's no mention of another recent work combining Wasserstein distances and optimal transport to RL by Terpin et al. (2020, below), which seems to be related.

### Experiments
* No description of the functional prior $p_0(f)$ used for the experiments was given. Was it a Gaussian process? How was it parameterised?
* I didn't find a description of what the shaded areas correpond to in the plots? Are they std. deviations due to multiple runs? If so, how many runs?

### Proof of Theorem 1
* There's a small mistake in the proof of Theorem 1, which doesn't seem to affect the final result. The third expectation in the second line of Eq. 20 should be on the negative log-prior $- \log p_0(f)$.
* It's a bit confusing to use log probability densities in the proof, as densities (w.r.t. the Lebesgue measure) are not defined in infinite-dimensional function spaces. The derivations are probably valid for a finite-dimensional truncation of $f$, which could possibly be extended to the infinite-dimensional setting by taking limits or a supremum. This is possibly a minor technicality, which might not affect the result, but it should be addressed or properly discussed to make the proof more solid and ensure it's valid for the settings that this paper considers.
* What particular result in Otto and Villani [2000] is the second inequality in Theorem 1 based on? That paper has many theoretical results. The citation should be more specific.
* How is the Talagrand inequality satisfied for a general $q(f)$?

### Minor issues
* The text has quite a few writing issues, which require a thorough revision. Some examples are listed below.
    - Typos: "(...) BNN is only used in weight space by the exciting [existing?] works (...)"
    - Citation formatting: A few in-text citations are inappropriately formatted as parenthetical citations, and vice-versa. Examples:
        - "(...) and function-space can be found in [Williams and Rasmussen, 2006]" -- In this case, the order of the author names is also swapped.
        - "generalized variational inference (or more general Rule of Three) Knoblauch et al. [2019]"
    - Choice of word: "... hyperparameters which [whose] details can be found in the Supplementary"
    - Missing articles: In general, a few terms are missing definite (e.g., *the*) or indefinite (e.g., *a/an*) throughout the text, making some references possibly ambiguous.
* Confusing sentences:
    - "Theorem 2. Let an old policy is..."
    - "used amortize fashion to resolve KL" - amortize fashion?

### References
* Terpin, A., Lanzetti, N., Yardim, B., Dorfler, F., & Ramponi, G. (2022). Trust region policy optimization with optimal transport discrepancies: Duality and algorithm for continuous actions. Advances in Neural Information Processing Systems, 35, 19786-19797.

**Q9 Complying With Reviewing Instructions:**

Yes

---

> ### Author Rebuttal · Authors · 2024-04-05
>
> ## Thanks for your detailed comments and support!
> > Theorem 1
>
> **Reply**: Thanks for the comment! The condition about $\log p_0(f)$ is already mentioned in the Theorem, and we will add the Talagrand's inequality condition to the Theorem in the final version. As for Lipschitz condition on $\log p_0(f)$, we can see that the finite counterpart (i.e., Gaussian distribution) satisfies this condition because its log probability density function is a polynomial function that satisfies the Lipschitz condition. In our implementations, we always use the measurement set and then $p_0(f)$ defined on this set would be a Gaussian distribution, so it is safe to evaluate the Wasserstein distance. As for the infinite case, there is no conclusion in the literature that the log functional probability density of GP would satisfy the Lipschitz condition or not because there is no explicit analytic form of its `functional probability density'. However, considering its finite counterpart, we might guess that it is possible for GP to satisfy such a condition as well because GP is sometimes considered as a vector (with infinite length) Gaussian distribution (not a firm conclusion). We will consider its proof in our future work and will identify this potential issue in the Conclusion.
>
> We define $p(D|f) = \exp{(J(D, f))}$ where $D$ denotes the observed episodes and $J(D, f)$ could be any surrogate term, and $p(D) =  \int_f p(D|f) \rm{d} f = \int_f \exp{(J(D, f))} \rm{d} f$. The above definitions are similar to the MEPO but with a function rather than a vector as the latent variable. Note that the reviewer may be concerned that the `likelihood' here is not a well-specific probability. In fact, it can be any cost function of latent variable and observation in the generalized variational inference framework [1].
>
> > Final formulation (Eq 12)
>
> **Reply**: Three terms are introduced one by one (i.e., Eqs (8), (9), and (11)) before the final Eq (12). We will add more transition texts before (12) to describe the links between the three terms and the motivation to add them together.
>
> About the monotonic improvement property, we want to highlight that the additional two terms are independent of the old policy $q_{old}(f)$, so the general improvement trend would be preserved during the training. However, the new policy needs to comprise three constraints/terms, so they may bring `noises' to the improvement trend at each step like the stochastic gradient optimization.
>
> The hyper-parameters are selected to ensure that the three terms are with similar scales in our implementation. We do believe there are some better ways to determine these hyper-parameters, like using some sophisticated black-box optimization methods, but we did not use them in our implementation.
>
> > Experiments
>
> **Reply**: Since we have no prior knowledge about the targeted problem, the prior used in the experiments is just a BNN with the same architecture of the policy network after training with a small number of episodes following [Rudner, et al, NIPS2022]. In practice, if we have some knowledge about the underlying policy function characteristics, we can express such knowledge using GP and replace BNN prior easily. We just need the prior to have the ability to obtain function samples from measurement sets.
>
> We use a fixed seed for one run in all experiments. The shade in the figure is the std of rewards accumulated in a 5-step window. We will give such details in the updated version.
>
> > proof of Theorem 1
>
> **Reply**: Yes, you are right about the log probability densities $\log p(f)$ and we just follow [Tran, et al, NIPS 2022] to use this expression. We do agree with you that it would be better to have solid proof of this density, but the difficulty of proving or investigating this density is expected to exceed this work so we will consider it as our future work.
>
> About Talagrand inequality result, it is from Definition 1 of the paper [2] which further sources from Theorem 1.1 of the paper [3] (Note that when the cost function is not with a square power, the inequality should be Eq (1.2) of [3]). The condition of this inequality is that the probability measure $p$ is absolutely continuous w.r.t. $q$ with finite moments of order 2. Such a condition is mainly used to ensure the KL divergence is well-defined, so it is only a mild condition for general situations.
>
> > other minor issues
>
> **Reply**: Thanks! We will correct all issues in the final version.
>
> &nbsp;
>
> ## References
> [1] Knoblauch, Jeremias, et al. "An optimization-centric view on Bayes' rule: Reviewing and generalizing variational inference." JMLR  132 (2022): 1-109.
>
> [2] Otto, Felix, and Cédric Villani. "Generalization of an inequality by Talagrand and links with the logarithmic Sobolev inequality." Journal of Functional Analysis 173, no. 2 (2000): 361-400.
>
> [3] Talagrand, Michel. "Transportation cost for Gaussian and other product measures." Geometric \& Functional Analysis GAFA 6, no. 3 (1996): 587-600.

---

### Official Review · Reviewer_4TYv · 2024-03-26

**Q2-1 Originality-Novelty:** 3
**Q2-2 Correctness-Technical Quality:** 3
**Q2-5 Clarity Of Writing:** 4

**Q1 Summary And Contributions:**

- introduction of a novel method for policy optimization : replace KL(old policy | new policy) regularization term of TRPO with Wasserstein distance of old and new policy in function space
- show that the regularization term represents a tighter variational bound than TRPO (might be sketchy)
- some empirical evaluations on CartPole and MuJoCo benchmarks

**Q2-3 Extent To Which Claims Are Supported By Evidence:**

3: Good: the main claims are supported by convincing evidence (in the form of adequate experimental evaluation, proofs, (pseudo-)code, references, assumptions).

**Q2-4 Reproducibility:**

4: Excellent: key resources (e.g. proofs, code, data) are available and key details (e.g. proof sketches, experimental setup) are comprehensively described for competent researchers to confidently and easily reproduce the main results.

**Q3 Main Strengths:**

- paper is well written, easy to follow and connects well to related work
- can follow the information flow
- hence proposed method seems mostly logical + straight forward
- experimental evaluation on most claims:
  - show your method less sensitive to initialization OK
  - show improved environment generalisation OK
  - show importance of uncertainty via environment uncertainty OK

**Q4 Main Weakness:**

Experimental evaluation:

1 - I believe the claim that the KL is a problematic regularizer for all the reasons the authors mention but if the claim that function space is better than weight space is not as convincing to me (for practical purposes)
-> ideally show experimental evaluation for that claim  : run Wasserstein regularizer in weight and function space

2 - show cases where KL collapses, stuck in local optimum

**Q5 Detailed Comments To The Authors:**

Questions and Comments:

- claim in abstract: "weight space modeling limits uncertainty modeling ..." -> Can you justify that further, I am not sure why that is true.
- I don't see how free lunch what you describe after please clarify that
- in BNN could mention local re-parameterization for variance reduction
- in section 3 extend the justification for your approach why does weight space reduce function flexibility (for nearly infinitely large networks? I am not sure that is true for relevant situations, like with large # of params we get an almost infinite feature space)
- Theorem 1: I am not sure I understand the proof. if the condition is fulfilled than the L^W >= L^KL, correct? And only than the bounds are true but generally not. Hence the claim that it is always a tighter bound is not true only if that condition is met. (If I am right pelase correct me if not. I would tone that down a lot Theorem -> Comment)
- typo in theorem 2
- Can you make the figure texts more informative?

**Q9 Complying With Reviewing Instructions:**

Yes

---

> ### Author Rebuttal · Authors · 2024-04-05
>
> ## Thanks for your comments and great support!
>
> > ideally show experimental evaluation for that claim: run Wasserstein regularizer in weight and function space
>
> **Reply**: In Figure 2, we have included the comparison between weight-space Wasserstein regularizer (BNN-W-PPO) and function-space Wasserstein regularizer (fBNN-W-PPO), where the functional one performs better than the weight-space one.
>
> > show cases where KL collapses, stuck in local optimum
>
> **Reply**: In Figure 1, we have shown that KL divergence will stuck in local optimum (the one closest to the initialization position) but the Wasserstein distance does not.
>
> > claim in abstract: "weight space modeling limits uncertainty modeling ..." -> Can you justify that further, I am not sure why that is true.
>
> **Reply**: The reason is that the samples under i.i.d Gaussian weight priors, tend to be horizontally linear for deep nets (as shown in Figure 1 of [3]) and the function posterior will bias to that horizontally linear function as well. Such pathological issue of weight-space prior would not only lead to limited function modelling capability but also limited uncertainty estimation capability.
>
> > I don't see how free lunch what you describe after please clarify that
>
> **Reply**: In the third paragraph of the Introduction, we want to highlight that although introducing BNN as policy parameterization could increase the uncertainty modelling ability of policy optimization, it does bring additional challenges to posterior inference. Due to the used i.i.d Gaussian priors, the faced challenges include: 1) it is hard to resolve the horizontally linear function bias from such prior; and 2) the effect of the prior on the functional output is not obvious to characterize and control [3].
>
> > in section 3 extend the justification for your approach why does weight space reduce function flexibility (for nearly infinitely large networks? I am not sure that is true for relevant situations, like with large # of params we get an almost infinite feature space)
>
> **Reply**: The reason is that, as we know, the BNN with infinite width would correspond to a GP [1,2], so any practical BNNs (with finite width) would have less function flexibility compared with the infinite width one (and its corresponding GP as well). If you could define an infinitely large BNN, such BNN would have the same (not less) function flexibility as GP, but we normally could only define a finite one.
>
>
>
> > Theorem 1: I am not sure I understand the proof. if the condition is fulfilled than the L^W >= L^KL, correct? And only than the bounds are true but generally not. Hence the claim that it is always a tighter bound is not true only if that condition is met. (If I am right please correct me if not. I would tone that down a lot Theorem -> Comment)
>
> **Reply**: Yes, the claim of the theorem is under the condition. Firstly, we did not state that this claim is "always" right. The theorem can only guarantee the correctness of this claim when the condition is met. If this condition is not met, we are unsure about the correctness of this claim. Secondly, we do not think it is a problem because 1) the condition is easy to meet since the prior is normally given by experts (like ourselves) in advance of the model training; 2) As far as we know, most (if not all) of the theorems in this area are with some conditions, which does not underestimate their valuable insights to the problems.
>
> > Other comments about typo, figure texts, and local re-parameterization
>
> **Reply**: Thanks for the comments and suggestions! We will update them in the final version.
>
> &nbsp;&nbsp;
>
> ### References
>
> [1] Lee, Jaehoon, Yasaman Bahri, Roman Novak, Samuel S. Schoenholz, Jeffrey Pennington, and Jascha Sohl-Dickstein. "Deep neural networks as gaussian processes." arXiv preprint arXiv:1711.00165 (2017).
>
> [2] Garriga-Alonso, Adrià, Carl Edward Rasmussen, and Laurence Aitchison. "Deep convolutional networks as shallow gaussian processes." arXiv preprint arXiv:1808.05587 (2018).
>
> [3]. Tran, Ba-Hien, Simone Rossi, Dimitrios Milios, and Maurizio Filippone. "All you need is a good functional prior for Bayesian deep learning." Journal of Machine Learning Research 23, no. 74 (2022): 1-56.

---

### Meta-Review · Area_Chair_Xsfd · 2024-04-17

While the reviewers didn't reach a consensus, the authors provided comprehensive responses to the reviewers' comments.The paper provides an important contribution to the policy optimization literature.